# EXPLORING CROSS-MODAL FLOWS FOR FEW-SHOT LEARNING

**Ziqi Jiang, Yanghao Wang, and Long Chen**[†]
Department of CSE, The Hong Kong University of Science and Technology
`zjiangbl@connect.ust.hk, ywangtg@connect.ust.hk, longchen@ust.hk`

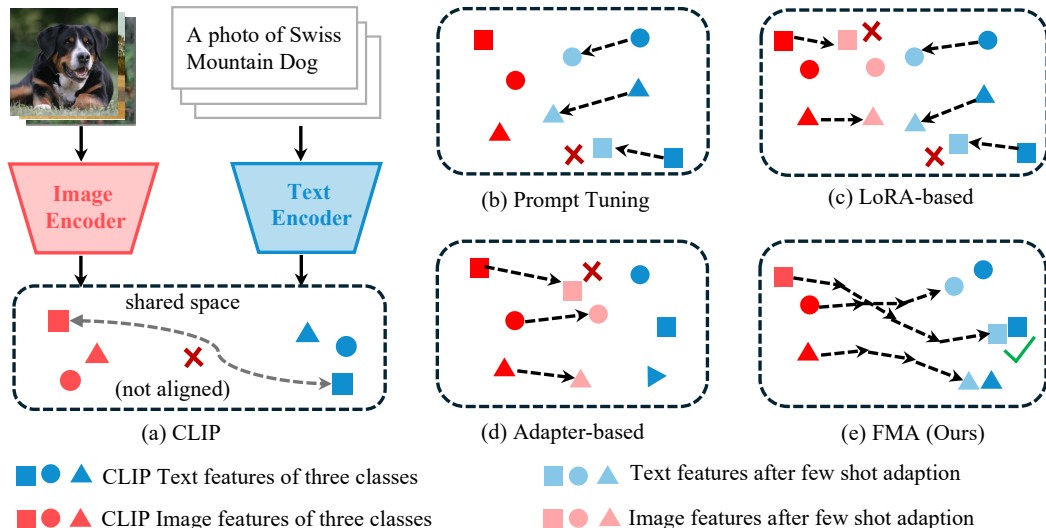

Figure 1: **Comparisons of the cross-modal alignment process of different methods**. **(a)** The overview pipeline of CLIP (Radford et al., 2021) for zero-shot cross-modal alignment and classification. Some image features and their corresponding text features are not well-aligned. **(b-d)** The alignment process of three typical types of state-of-the-art PEFT approaches, which adjust image or text features in *one single step*. The arrow shows the adjustment of corresponding features during the adaptation. For difficult classes, the image features still may be far from the corresponding text features by one step adjustment. **(e)** FMA achieves *multi-step* cross-modal alignment, which succeeds in aligning text-image features for difficult classes.

## ABSTRACT

Aligning features from different modalities, is one of the most fundamental challenges for cross-modal tasks. Although pre-trained vision-language models can achieve a general alignment between image and text, they often require parameter-efficient fine-tuning (PEFT) for further adjustment. Today's PEFT methods (*e.g.*, prompt tuning, LoRA-based, or adapter-based) always selectively fine-tune a subset of parameters, which can slightly adjust either visual or textual features, and avoid overfitting. In this paper, we are the first to highlight that all existing PEFT methods perform *one-step adjustment*. It is insufficient for complex (or difficult) datasets, where features of different modalities are highly entangled. To this end, we propose the first model-agnostic *multi-step adjustment* approach by learning a cross-modal velocity field: Flow Matching Alignment (**FMA**). Specifically, to ensure the correspondence between categories during training, we first utilize a fixed coupling strategy. Then, we propose a noise augmentation strategy to alleviate the data scarcity issue. Finally, we design an early-stopping solver, which terminates the transformation process earlier, improving both efficiency and accuracy. Compared with one-step PEFT methods, FMA has the multi-step rectification ability to achieve better alignment. Extensive results demonstrate that FMA can yield significant performance gains across various benchmarks and backbones, particularly on challenging datasets.

---

[†]Long Chen is the corresponding author. Codes: https://github.com/HKUST-LongGroup/FMA

# 1 INTRODUCTION

How to align information from different modalities (*e.g.*, text and images), is very important in almost all cross-modality tasks. Well-aligned cross-modality features play a crucial role in a variety of domains, such as achieving the impressive reasoning abilities in MLLMs (Hurst et al., 2024; Li et al., 2022; Liu et al., 2023b), and realistic generation qualities in AIGC models (Rombach et al., 2022; Esser et al., 2024). Generally, realizing good cross-modal alignment means finding a "golden" multimodal distribution where corresponding text and image features are as close as possible in the shared common space. To achieve this goal, many pretrained vision-language models (VLMs) have been proposed, such as CLIP (Radford et al., 2021) and ALIGN (Jia et al., 2021). Taking CLIP for example, as shown in Figure 1(a), it achieves cross-modal alignment by training image and text encoders jointly with a contrastive objective. By now, VLMs can achieve a general alignment between image and text, and show promising results on zero-shot tasks, like image recognition.

However, due to the internal complexity of different modalities, pre-trained VLMs cannot achieve perfect alignment across all scenarios. Therefore, they typically require further fine-tuning steps to rectify features for better alignment. For instance, in few-shot learning, we usually need to fine-tune VLMs with a few images from base categories. Since fine-tuning the whole VLM model is computationally expensive, numerous parameter-efficient fine-tuning (PEFT) methods have been proposed. These PEFT methods can be broadly categorized into three groups: 1) **Prompt Tuning** (Li & Liang, 2021): Like CoOp (Zhou et al., 2022b) and CoCoOp (Zhou et al., 2022a), prompt tuning methods try to find better text representations (or features) by replacing handcrafted text prompts with learnable continuous prompts. Compared with original VLMs, they can be interpreted as applying a "movement" on all text features to better align them with the corresponding im-

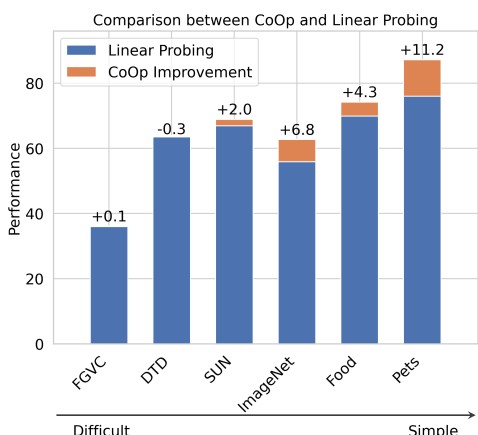

Figure 2: **Performance of CoOp and linear probing.** We experimented with the 16-shot setting and chose CLIP RN50 as the backbone.

age features (*cf.*, Figure 1(b)). 2) **Adapter-based** (Houlsby et al., 2019): Such as CLIP-Adapter (Gao et al., 2024), they typically apply a learnable adapter following the VLMs' image encoder. After training, they will rectify the image features towards corresponding text features for better alignment (*cf.*, Figure 1(c)). 3) **LoRA-based** (Hu et al., 2022): These methods (*e.g.*, CLIP-LoRA (Zanella & Ben Ayed, 2024)) add trainable low-rank matrices in both text and image encoders to store the new knowledge while keeping other parameters frozen. As shown in Figure 1(d), they will shift both text and image features to be closer (*i.e.*, towards the golden distribution).

Generally, the performance gains of PEFT methods stem from two sources: 1) the incorporation of few-shot training data, and 2) the algorithmic capability to align cross-modal features. To evaluate the latter, we utilize linear probing as a reference baseline. Since linear probing can only learn a simple linear boundary, it effectively represents the performance gain attributed solely to data access. Consequently, by measuring the margin by which PEFT outperforms linear probing, we can factor out the data's contribution and evaluate PEFT's cross-modal alignment capability. However, we observed that their advantages compared with linear probing are not consistent across different datasets. To better illustrate this, we introduce the concept of dataset difficulty: A dataset with lower CLIP RN50 zero-shot performance is considered more difficult. This definition is reasonable because lower zero-shot performance means that the cross-modal distribution is more complicated, thus more difficult to adjust. In this paper, we found that while PEFT methods work pretty well on relatively simple datasets, they fail to generalize to difficult ones. For example, as shown in Figure 2, the prevailing PEFT method CoOp can obtain remarkable improvements over the linear probing baseline on relatively simple datasets, like OxfordPets. However, the improvements on more challenging datasets, such as FGVCAircraft, are marginal.

In this paper, we attribute this limitation to the highly entangled cross-modal distributions in difficult datasets, which necessitate complex transformations for effective alignment. Existing PEFT methods struggle to model such transformations, primarily because they rely on a **single forward pass for**

**feature adjustment during inference** (we term these PEFT paradigms as *one-step* approaches). This forces the model to predict the target features directly from the input in one step, which is inherently difficult for very complicated cross-modal distributions. Since one-step approaches are difficult to model complex transformations, a natural question is: **Can we make *multi-step* adaptation to realize better cross-modal alignment?**

To address this, we turn to the Flow Matching (FM) theory (Liu et al., 2022; Lipman et al., 2022; Albergo & Vanden-Eijnden, 2022), a framework predominantly used in image generation. Generally, FM learns a velocity field that transports samples from a source distribution (typically prior noise) to a target distribution (e.g., real images) through a multi-step iterative process. The core motivation for this paradigm is that while directly mapping a noise into a real image is challenging(like GAN,VAE), **decomposing the transformation into multiple intermediate states make the problem more tractable**. At each step, the velocity field only needs to predict a local update based on the current state, which is significantly simpler than directly learning the one-step mapping. Although typically applied to noise-to-data generation, FM theoretically support transformations between any two arbitrary distributions. Therefore, if a velocity field can be trained to transform image features to corresponding text features, we can utilize its multi-step ability to align the complicated cross-modal distribution. Specifically, we can formulate all encoded image features as the source distribution. For the target distribution, the simplest method is to encode all prompts with category names as target features. For each velocity training step, an image and text feature will be sampled independently to compose a training pair. This velocity field will transform any given image feature (source distribution) to a corresponding text feature (target distribution), *i.e.*, classification.

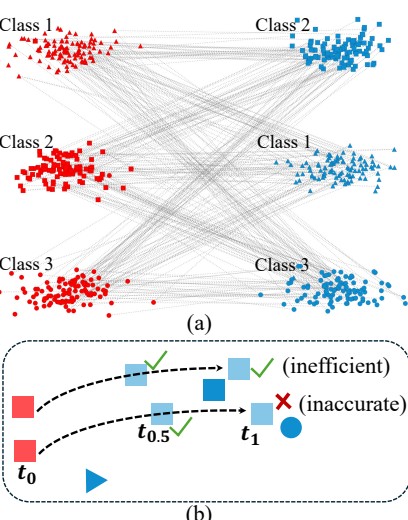

Figure 3: **Explanation of the two challenges.** (a) A flow matching example to transform image features (red) to text features (blue) distribution. Each distribution consists of features of three categories. (b) Two examples of how a trained velocity transforms an image feature into a text feature using the vanilla flow matching inference strategy.

Unfortunately, there are two potential issues for this straightforward approach. Firstly, although a well-trained velocity field can achieve transformation between two given distributions, it is not guaranteed to preserve local structure (*e.g.*, category correspondence). For example, as shown in Figure 3(a), it may transform image features from one class to text features of another class, resulting in wrong classification. Therefore, **the first challenge** is to figure out *how to train a velocity field, which can transfer image features from source distribution, not only to the target distribution (near text features), but also close to their right class embeddings.* Secondly, in the inference stage, the goal of flow matching for the generation task is to generate features in the target distribution, which are then decoded to real samples. But for the classification task, what we really need is to transfer image features closer to positive embeddings (*i.e.*, ground-truth class embeddings) than other negative ones. This inconsistency suggests that the previous flow matching inference strategy may not be the best practice for classification scenarios. Thus, **the second challenge** is that *how to design an appropriate inference method for classification.*

To overcome both challenges, we propose a novel framework for few-shot learning: Flow Matching Alignment (**FMA**). Specifically, we have three designs:

- **Coupling Enforcement:** Firstly, for any image feature, we choose the one corresponding to its class label to compose a training pair. Theoretically, this strategy can guarantee that the trained velocity field will act like a classifier, moving image features towards the correct direction.

- **Noise Augmentation:** However, compared with randomly pairing, this coupling strategy also reduces the number of training pairings dramatically, making it harder to train the velocity field. To solve this issue, we further add a pre-defined noise to the sampled pair during training. This augmentation makes the training data not constrained in a low-dimensional manifold, making the learning process more stable and robust.

- **Early Stopping Slover:** Besides training, vanilla flow matching inference is not suitable in classification due to the inconsistency (challenge #2). When intermediate features are distinguishable

enough for classification, continuing to transfer them to obtain text features is unnecessary and inefficient. Even worse, we observed that the following inference steps are likely to move them to inaccurate text features, as shown in Figure 3(b). This inaccuracy is because even with the previous training strategies, it is still difficult to train a perfect velocity field. To tackle this, we devise an early stopping solver for few-shot learning. Concretely, instead of arriving at the target distribution. We allow the model to output intermediate features for classification. This early stopping strategy can not only reduce the inference time when the features are good enough for classification, but also reduce the risk of transferring to wrong text features.

Additionally, FMA can serve as a plug-and-play module for mult-step rectification. It only requires two sets of features in the shared space, and it is agnostic to the specific methods used for image or text feature extraction. Thus, FMA can be easily adapted to different methods, ranging from zero-shot CLIP to various PEFT methods. To demonstrate the effectiveness and generality of FMA, we have integrated it with various backbones. Extensive results on different backbones and benchmarks have demonstrated consistent performance gains. In summary, our contributions are threefold:

- We propose a new perspective to analyze existing few-shot PEFT approaches: all these methods are one-step methods, and they usually struggle with difficult datasets.

- We are the first to explore the potential to adapt the multi-step recitification ability of flow matching for better few-shot learning performance. Based on this, we propose FMA, a novel plug-and-play multi-step rectification framework.

- Extensive results demonstrate the superiority and robustness of FMA in the few-shot classification.

## 2 RELATED WORK

**Few-Shot Learning in VLMs.** Few-Shot Learning (FSL) is a task to learn from a few examples (Wang et al., 2020; Yue et al., 2020). Currently, the most common scenario in FSL is to fine-tune a pre-trained VLM (Radford et al., 2021; Jia et al., 2021) with limited data for classification. Because directly fine-tuning VLMs is computationally expensive, numerous PEFT methods have been proposed. Current techniques for adapting the VLMs can be broadly classified into four paradigms: Prompt Tuning, Adapter-Based and LoRA-Based. The Prompt Tuning approaches (Zhou et al., 2022a;b; Lee et al., 2023; Bulat & Tzimiropoulos, 2023) involves learning textual or visual prompts that are subsequently inserted into the input or middle layers of the pre-trained VLM encoders for a few-shot adjustment. LoRA-Based methods (Zanella & Ben Ayed, 2024; da Costa et al., 2023; Kim et al., 2024; He et al., 2022), instead, insert a small number of trainable parameters (*e.g.*, LoRA ) within the encoders themselves. Adapter-based methods (Zhang et al., 2022; Gao et al., 2024), append a trainable layer like MLP to the frozen image or text encoders, and they do not require gradient backpropagation across the encoders, which is more computationally friendly.

**Diffusion Model (DM) and Flow Matching (FM).** Recently, diffusion model has developed into the most powerful framework in generative models (Ho et al., 2020; Song et al., 2020a). It gradually degrades the data by adding noise, then learn the reverse process. Then Score-SDE (Song et al., 2020b) points out that the learning process of DM is actually solving stochastic differential equations (SDE), which can be further interpreted as probabilistic ordinary differential equations (ODE). Consequently, FM (Liu et al., 2022; Lipman et al., 2022; Albergo & Vanden-Eijnden, 2022) extends this by learning a velocity field to build transformations between any two distributions. Different from previous one-step generative models, such as GAN (Goodfellow et al., 2014) and VAE (Kingma & Welling, 2013), FM generates data through multi-step rectification. So far, FM has gained great success in a variety of domains, such as text-to-image generation (Esser et al., 2024; Geng et al., 2025; Albergo et al., 2023), image editing (Kim et al., 2025; Kulikov et al., 2024; Rout et al., 2024), video generation (Jin et al., 2024; Polyak et al., 2024), and so on. However, although FM can achieve transformation between two arbitrary distributions, most approaches tend to set one as the prior noise distribution. Recently, several works (He et al., 2025; Liu et al., 2025) try to transform from text distribution to image distribution directly for generation. In this paper, we explore the potential of whether FM is suitable for supervised tasks like few-shot classification.

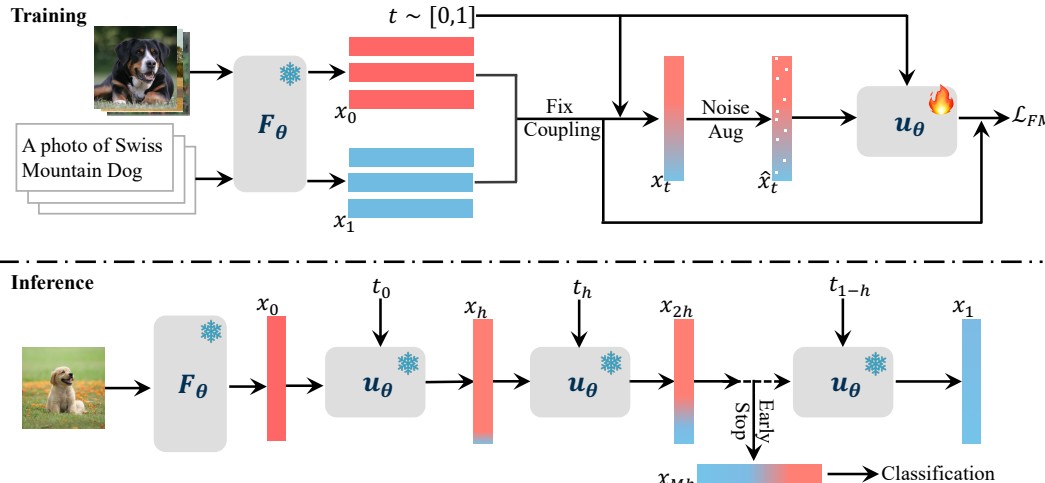

Figure 4: **Overview of Flow Matching Alignment (FMA)**. The main idea of the training stage is to learn a velocity field, which can transform image features to corresponding text features. Two designs are proposed: coupling enhancement and noise augmentation. During the inference, FMA applies an early-stopping solver that can output intermediate features for classification.

## 3 METHOD: FLOW MATCHING ALIGNMENT FOR FEW-SHOT LEARNING

**Formulation.** We consider the $N$-way $K$-shot classification problem: Given a training dataset consisting of $N$ classes, each class with $K$ examples, the goal is to train a classification model that can accurately predict the class of any test image from these $N$ classes.

**General Framework**. The overall pipeline of our proposed FMA is shown in Figure 4. Specifically, FMA consists of three steps: 1) Encoding all text and images into features in the shared common space. 2) Training a velocity field to transform image features to text features for better alignment. 3) Using the trained velocity field to transform the feature of the given test image for classification.

### 3.1 FEATURE EXTRACTION

Given the training dataset $\mathcal{D} = \{I^i, C^i\}_{i=1}^{N \cdot K}$, where $I^i, C^i$ represent an image and its corresponding class label, the simplest way to encode them into the shared space is using a pre-trained VLM $F_\phi$, like CLIP. Specifically, CLIP contains an image encoder $E_I$ and a text encoder $E_T$. The image (source) distribution is obtained by encoding all images into a feature representation by: $x_0 = E_I(I) \in \mathbb{R}^d$ where $d$ is the dimension of the feature. Here we omit the superscript for simplicity. For the text (target) distribution, CLIP first transforms all label $C$ into text descriptions $T$ using their class name and a handcrafted template like "a photo of {class}". Then the text encoder will encode these descriptions into text features by: $z = E_T(T)$, with the same dimension as the image feature. Notably, our method is agnostic to the feature extraction process. While we use CLIP zero-shot as an example, other methods like CoOp can also be used to extract image and text features. In the experiment section, we discuss the influence of different feature extraction methods on FMA.

After obtaining all image and text features, FMA trains a velocity field to perform further alignment. For simplicity, we denote distributions $p_0$ and $p_1$ as the collections of image features and text features.

### 3.2 TRAINING STAGE

Typically, in flow matching training, we **randomly** sample an image feature $x_0 \sim p_0$ and a text feature $z \sim p_1$ to compose a training pair. Following Rectified Flow (Liu et al., 2022), a transfer trajectory is defined as linear interpolation between $x_0$ and $z$ on time $t \in [0, 1]$: $x_t = tz + (1 - t)x_0$. The conditional velocity field $v_t(x_t|z)$ is defined as the derivative of the trajectory with respect to time $t$: $v_t(x_t|z) = \frac{\mathrm{d}x_t}{\mathrm{d}t} = z - x_0$. The marginal velocity field $u_t^\theta(x_t)$ can be learned by solving a

simple least square regression problem between $v_t(x_t|z)$ and $u_t^\theta$ :

$$\mathcal{L}_{FM}(\theta) = \mathbb{E}_{x_t \sim p_t, t \sim [0,1]}[\|u_t^\theta(x_t) - (z - x_0)\|^2], \qquad (1)$$

where we use the distribution $p_t$ to denote the collection of all intermediate features $x_t$. However, the learned velocity field learned $u_t^\theta(x_t)$ cannot guarantee class-level correspondence because it is an estimation of $v_t(x_t)$, an expectation of the $v_t(x_t|z)$ over all possible text features $z$:

$$v_t(x_t) = \int v_t(x_t|z) \frac{p_t(x_t|z)p_1(z)}{p_t(x_t)} dz. \qquad (2)$$

Consequently, the learned $u_t^\theta$ will drive an image feature towards the average of all text features, which usually results in incorrect classification.

**Coupling Enforcement.** To solve this issue, we propose coupling enforcement. Specifically, for a given image feature $x_0$, we **exclusively** sample its corresponding ground-truth text feature $z_c$. Due to the sparsity in the high-dimensional manifold, their transfer trajectories can be considered as mutually non-crossing when the dataset is small. This means for given $x_t$, there exists only one potential text feature $z$ in Eq. (2). Consequently, the marginal velocity is equivalent to the conditional one:

$$v_t(x_t) = v_t(x_t|z_c). \qquad (3)$$

This means we are secretly using the conditional velocity field $v_t(x_t|z_c)$ to move the image feature to the direction of $z_c$. Because $z_c$ corresponds to the right category, $v_t(x_t)$ can then transform the image feature to the correct text feature in the target distribution. For more details, refer to Section 4.

While this coupling enforcement strategy is intuitive, it induces another new challenge: data scarcity. Since we only pair one image feature with its target text feature, the model is trained on only $\frac{1}{N}$ of the potential data compared with randomly pairing, where $N$ is the number of classes in the dataset. As a result, large portions of the velocity field's domain remain unsampled, preventing it from providing reliable instructions to move image features.

**Noise Augmentation.** We design another strategy, noise augmentation, to solve this problem. It injects a time-dependent Gaussian noise into the intermediate features $x_t$ during the training process. Adding random noise ensures the distribution $x_t$ does not collapse to a low-dimensional manifold (Song & Ermon, 2019), thus learning more accurate velocity estimation. Concretely, the noise schedule is chosen as a time-dependent sequence. Inspired by Schrödinger bridge (Liu et al., 2023a), we sample the new noise-augmented features $\hat{x}_t$ from a Gaussian distribution:

$$\hat{x}_t \sim \mathcal{N}\left(\hat{x}_t|x_t, t \cdot (1-t) \cdot \sigma^2(x_t)\right). \qquad (4)$$

Here $\sigma^2(x_t) \in \mathbb{R}$ is the standard deviation of $x_t \in \mathbb{R}^d$: $\sigma^2(x_t) = \sqrt{\sum_{i=1}^{d}(x_t^i - \mu(x_t))^2/d}$, where $\mu(x_t) = \sum_{i=1}^{d} x_t^i/d$ is the mean of $x_t$. $x_t^i \in \mathbb{R}$ is the i-th dimension of $x_t$. After obtaining $\hat{x}_t$, the new ground truth $u_t^\theta(\hat{x}_t)$ is the direction pointing to the target text feature $z_c$: $v_t(\hat{x}_t|z_c) = \frac{z_c - \hat{x}_t}{1-t}$. Then Eq. (1) can be applied for training $u^\theta(x_t)$. The training is summarized in Algorithm 1.

### 3.3 INFERENCE STAGE

After learning a velocity field $u_t^\theta(x_t)$, we use it to transform $x_0$ for classification. Vanilla flow matching inference process adapts an ODE solver, such as the Euler Method, to iteratively transform $x_0$ into a text feature $x_1$ by $M$ steps: $x_{t+h} = x_t + h \cdot u_t^\theta(x_t)$. Here $h = \frac{1}{M}$ is the integration stepsize. $x_1$ is an approximation of a sample from $p_1$ and used for classification: Calculate its cosine similarity with the text features of all classes, then select the class with the highest similarity as the prediction.

However, this inference method may lead to an inconsistency issue, as we mentioned before. Specifically, as shown in Figure 5(a), with $t$ approaching 1, the distance between intermediate features $x_t$ and corresponding text features becomes smaller, which means they are indeed moving towards to the target distribution. However, the classification accuracy using $x_t$ first increases, then decreases. This indicates that although the coupling enhancement strategy is adapted, some features may still move towards text features of incorrect classes. To better illustrate this, we show an example in Figure 5(b) where this inference method leads to a classification error. As the transformation continues, $x_t$ becomes closer to the incorrect text feature than to the correct one. In this situation, the vanilla inference method, which use $x_1$ for classification, will lead to incorrect results.

**Early Stopping Solver (ESS).** To solve this problem, We propose a simple yet effective inference strategy: utilizing an early stopping solver (ESS) to terminate the transformation when the intermediate features are discriminative for classification. Specifically, instead of integrating over the full time interval $[0, 1]$, we fix the step-size $h$ as a constant and choose a constant number of inference steps $M$. $u_t^\theta(x_t)$ then transforms an initial image feature $x_0$ to an intermediate feature $x_{\hat{T}}$ at the final time $\hat{T} = h \cdot M$, as shown in Algorithm 2. Subsequently, $x_{\hat{T}}$ is used for classification. Notably, the optimal strategy is to find a sample-specific $t$ for each input image feature, and we left this as a promising direction for future research.

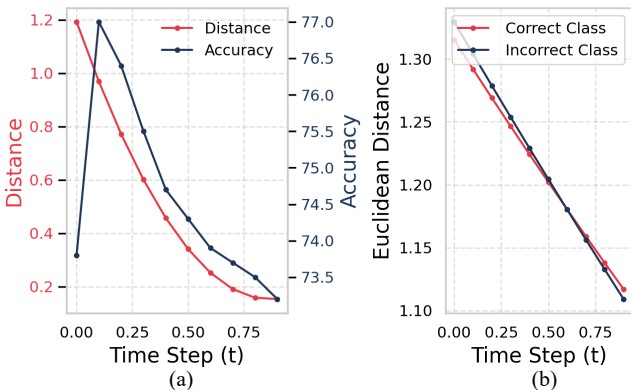

Figure 5: (a) Red line: Average distance to target figure at different timesteps. Black line: Accuracy using features at different timesteps for classification. (b) At different timesteps, the distance between the intermediate features and the correct/incorrect text feature.

## 4 THEORETICAL ANALYSIS

In this section, we provide a rigorous derivation of FMA. Given image (source) distribution $p_0$ and text (target) distribution $p_1$, FMA attempts to find a velocity $v_t(x_t)$, which can transform a given image feature $x_0 \in p_0$ to its **corresponding** text feature through: $\mathrm{d}x_t = v_t(x_t)\mathrm{d}t$.

Because the closed-form of $v_t(x_t)$ is infeasible to calculate, a neural network cannot be used to learn it directly. Therefore, we apply the conditional flow matching as a proxy task. Specifically, consider a text embedding $z \in p_1$, first we need to define its conditional probability path $p_t(x_t|z)$.

However, defining $p_t(x_t|z)$ in FMA is not as simple as that in generative flow matching. In generation, given any $z$, $p_0(x_0|z)$ is simply an identical prior distribution. Nevertheless, in FMA, the different $z$ represent different categories, yield different conditional probability paths $p_0(x_0|z)$. Therefore, how to define $p_0(x_0|z)$ is the first problem in FMA.

**Coupling Enhancement**. Actually, the Coupling Enhancement strategy is secretly defining a conditional probability path $p_0(x_0|z)$. Mathematically, it defines $p_0(x_0|z)$ as follow:

$$p_0(x_0|z) = \sum_{i=1}^{K} \delta(x_z^i). \tag{5}$$

Where $x_z^i$ means an image feature of the category $z$, and $\delta(\cdot)$ represents the Dirac function. Then $p_t(x_t|z)$ over time $t \in (0, 1]$ can be defined as a linear interpolation between $x_0$ and $z$:

$$p_t(x_t|z) = \delta(tz + (1-t)x_0) = \mathcal{N}(tz + (1-z)x_0, 0). \tag{6}$$

**Proposition 1** (Identity). *Consider an intermediate feature $x_t$, and $z_c \in p_1$ is its corresponding text feature. Then the marginal velocity field is identical to the condition velocity field of $x_t$:*

$$v_t(x_t) = v_t(x_t|z_c).$$

Where $v_t(x_t)$ is defined as Eq. (2) and $v_t(x_t|z_c) = \frac{\mathrm{d}x_t}{\mathrm{d}t} = z_c - x_0$. The proof is given in Appendix A.

**Proposition 2** (Classification). *Given any image feature $x_0$ and its text embedding $z_c \in p_1$, if a conditional velocity field $v_t(x_t|z_c)$ is defined to generate $p_t(x_t|z_c)$, and $v_t(x_t)$ is defined by Eq. (2). Then we have*: $x_1 = z_c$. *where $x_1$ is estimated by*: $\mathrm{d}x_t = v_t(x_t)\mathrm{d}t$.

This means $v_t(x_t)$ theoretically can guarantee that the transferred feature $x_1$ is exactly the correct text embedding $z_c$. The proof is given in Appendix A.

**Noise Augmentation**. This strategy can be viewed as an extension to Eq. (6) if we consider a non-zero variance in the conditional probability path $p_t(x_t|z)$, which will naturally yield the result in

Eq. (4). According to Zhu et al. (2025), a non-zero variance probability path (namely Schrödinger bridge) can augment the training trajectories, thus boosting the performance. However, it will also make the training procedure more difficult due to the random noise.

## 5 EXPERIMENTS

**Datasets and Models.** We evaluated FMA on the few-shot classification task. Specifically, We conducted experiments on 11 benchmarks, including Aircraft (Maji et al., 2013), DTD (Cimpoi et al., 2014),EuroSAT (Helber et al., 2019), SUN397 (Xiao et al., 2010), StanfordCars (Krause et al., 2013), ImageNet (Deng et al., 2009), UCF101 (Soomro et al., 2012), Flowers102 (Nilsback & Zisserman, 2008), Food101 (Bossard et al., 2014), OxfordPets (Parkhi et al., 2012), Caltech101 (Fei-Fei et al., 2004), sorted from difficult to easy.[1] To better illustrate the effect of dataset difficulty on our method, we divided these datasets into two groups: the first five as the difficult set, and the remaining six as the easy set. For each dataset, we followed the standard protocol to split the training, validation, and test sets. For $K$-shot classification, we randomly sample $K$ images from each class as the training set. Unless otherwise specified, the training epoch is set to 200. The architecture of the velocity network is the same as MAR (Li et al., 2024). By default, there are 6 ResNet blocks, and all hidden dimensions are equal to CLIP's feature dimension.

### 5.1 COMPARISONS WITH STATE-OF-THE-ARTS

**Settings.** We compared our FMA framework with 8 state-of-the-art methods, including CoOp (Zhou et al., 2022b), CoCoOp (Zhou et al., 2022a), CLIP-Adapter (Gao et al., 2024), Tip-Adapter (Zhang et al., 2022), PLOT++ (Chen et al., 2022), KgCoOp (Yao et al., 2023), ProGrad (Zhu et al., 2023) and CLIP-LoRA (Zanella & Ben Ayed, 2024). We implemented our FMA framework on top of CLIP-LoRA. Specifically, we first used a pre-trained CLIP-LoRA model to extract image and text features. After that, a velocity network was trained on these features according to FMA framework. Notably, the dataset for training CLIP-LoRA and the velocity field are kept the same, which means no extra knowledge is introduced.

Table 1: **Evaluation of FMA's generalization.** "D" and "E" mean average performance on difficult and easy datasets. Higher values between baseline and FMA are bolded.

| Method | Adaptation | | Generalization | | Harmonic | |
|---|---|---|---|---|---|---|
| | D | E | D | E | D | E |
| CLIP | 48.9 | 78.8 | 49.0 | **81.2** | 48.9 | 79.6 |
| +FMA | **68.9** | **87.6** | **49.4** | 81.0 | **57.5** | **84.2** |
| CoOp | 71.4 | 87.1 | 44.5 | **75.8** | 54.8 | 81.1 |
| +FMA | **74.0** | **87.9** | 44.5 | 75.7 | **55.6** | **81.3** |
| CoCoOp | 64.1 | 85.0 | 49.5 | 82.1 | 55.9 | 83.5 |
| +FMA | **68.5** | **87.3** | **50.6** | **82.2** | **58.2** | **84.7** |
| CLIP-Adapter | 62.4 | 86.6 | 47.0 | 79.8 | 53.6 | 83.1 |
| +FMA | **67.9** | **87.4** | **47.7** | **80.0** | **56.0** | **83.5** |
| CLIP-LoRA | 76.1 | 88.2 | 45.7 | 80.5 | 57.1 | 84.2 |
| +FMA | **77.8** | **88.8** | **46.7** | **80.6** | **58.4** | **84.5** |

The AdamW optimizer was used in FMA training with a learning rate of 0.0002, and we used ViT-B/16 backbone as the default setting. For FMA inference, we set the default stepsize $h = 0.1$. To find the optimal number of inference steps $M$, we first attempted various values in the validation dataset, then chose the one with the highest performance as the final number of inference steps.

**Results.** We reported the classification accuracy of all methods on 11 datasets in Table 2. For each dataset, we reported the results of 1-shot, 4-shot, and 16-shot classification. The results show that FMA achieves the best performance on most datasets. Notably, compared with simple datasets, FMA shows more significant effectiveness on difficult datasets. This validates our previous conclusion: Multi-step rectification is necessary when the cross-modal distribution is complicated to align.

### 5.2 GENERALIZATION ABILITY

**Settings.** While many PEFT approaches like CoOp can improve the performance of few-shot classification, they usually result in the degradation of the generalization ability. Therefore, we explored whether FMA will cause this problem by evaluating the cross-dataset transferability. Specifically, we first chose Imagenet to train the velocity field based on five different baselines. Then we tested the velocity on difficult and easy datasets, respectively. To conduct a more comprehensive evaluation, we

---

[1]According to the zero-shot performanc of CLIP ViT-B/16, see Table 11 in CLIP (Radford et al., 2021).

Table 2: **Comparison with other state-of-the-art approaches.** Based on CLIP-LoRA, we add FMA to further improve the performance. The highest value of each dataset is bolded.

| Shots | Method | Difficult | | | | | | Easy | | | | | | |
|---|---|---|---|---|---|---|---|---|---|---|---|---|---|---|
| | | Aircraft | SAT | DTD | SUN | Cars | Avg | UCF | Net | Flowers | Food | Pets | Caltech | Avg |
| 0 | CLIP (2021) | 24.8 | 47.8 | 43.8 | 62.5 | 65.5 | 48.9 | 66.7 | 66.7 | 67.4 | 85.3 | 89.1 | 92.9 | 78.0 |
| 1 | CoOp (2022b) | 20.8 | 56.4 | 50.1 | 67.0 | 67.5 | 52.4 | 71.2 | 65.7 | 78.3 | 84.3 | 90.2 | 92.5 | 80.4 |
| | CoCoOp (2022a) | 28.1 | 55.4 | 52.6 | 68.7 | 67.6 | 54.5 | 70.4 | 69.4 | 73.4 | 84.9 | 91.9 | 94.1 | 80.7 |
| | TIP-Adapter (2022) | 28.8 | 67.8 | 51.6 | 67.2 | 67.1 | 56.5 | 73.4 | 69.4 | 83.8 | 85.8 | 90.6 | 94.0 | 82.8 |
| | CLIP-Adapter (2024) | 25.2 | 49.3 | 44.2 | 65.4 | 65.7 | 50.0 | 66.9 | 67.9 | 71.3 | 86.1 | 89.0 | 92.0 | 78.9 |
| | PLOT++ (2022) | 28.6 | 65.4 | 54.6 | 66.8 | 68.8 | 56.8 | 74.3 | 66.5 | 80.5 | 86.2 | 91.9 | 94.3 | 82.3 |
| | KgCoOp (2023) | 26.8 | 61.9 | 52.7 | 68.4 | 66.7 | 55.3 | 72.8 | 68.9 | 74.7 | **86.4** | 92.1 | 94.2 | 81.5 |
| | ProGrad (2023) | **28.9** | 57.0 | 52.8 | 67.0 | 68.2 | 54.8 | 73.3 | 67.0 | 80.9 | 84.9 | 91.4 | 93.5 | 81.8 |
| | CLIP-LoRA (2024) | 28.0 | 71.9 | 54.1 | 70.3 | 69.4 | 58.7 | 75.4 | **70.3** | 81.4 | 85.1 | 91.9 | 93.8 | 83.0 |
| | +FMA (Ours) | 28.3 | **73.0** | **55.1** | **70.6** | **69.8** | **59.4**+0.7 | **75.9** | 70.2 | **84.9** | 85.2 | **92.1** | **94.5** | **83.8**+0.8 |
| 4 | CoOp (2022b) | 30.9 | 69.7 | 59.5 | 69.7 | 74.4 | 60.8 | 77.6 | 68.8 | 92.2 | 84.5 | 92.5 | 94.5 | 85.0 |
| | CoCoOp (2022a) | 30.6 | 61.7 | 55.7 | 70.4 | 69.5 | 57.6 | 75.3 | 70.6 | 81.5 | 86.3 | 92.7 | 94.8 | 83.5 |
| | TIP-Adapter (2022) | 35.7 | 76.8 | 59.8 | 70.8 | 74.1 | 63.4 | 78.1 | 70.7 | 92.1 | 86.5 | 91.9 | 94.8 | 85.7 |
| | CLIP-Adapter (2024) | 27.9 | 51.2 | 46.1 | 68.0 | 67.5 | 52.1 | 70.6 | 68.6 | 73.1 | 86.5 | 90.8 | 94.0 | 80.6 |
| | PLOT++ (2022) | 35.3 | 83.2 | 62.4 | 71.7 | 76.3 | 65.8 | 79.8 | 70.4 | 92.9 | 86.5 | **92.7** | 95.1 | 86.2 |
| | KgCoOp (2023) | 32.2 | 71.8 | 58.7 | 71.5 | 69.5 | 60.7 | 77.6 | 69.9 | 87.0 | **86.9** | 92.6 | 95.0 | 84.8 |
| | ProGrad (2024) | 34.1 | 69.6 | 59.7 | 71.7 | 75.0 | 62.0 | 77.9 | 70.2 | 91.1 | 85.4 | 92.1 | 94.4 | 85.2 |
| | CLIP-LoRA (2024) | 38.8 | 83.5 | 64.0 | 72.8 | 77.4 | 67.3 | 81.1 | 71.4 | 92.9 | 82.6 | 90.6 | 95.0 | 85.6 |
| | +FMA (Ours) | **40.3** | **85.0** | **67.0** | **73.7** | **78.9** | **69.0**+1.7 | **82.4** | **72.0** | **95.0** | 83.2 | 90.8 | **95.8** | **86.5**+0.9 |
| 16 | CoOp (2022b) | 43.3 | 86.0 | 70.0 | 74.9 | 83.1 | 71.5 | 83.1 | 71.4 | 97.2 | 84.4 | 91.1 | 95.5 | 87.1 |
| | CoCoOp (2022a) | 33.8 | 75.5 | 65.8 | 72.8 | 72.4 | 64.1 | 76.0 | 71.1 | 87.1 | 87.4 | 93.2 | 95.2 | 85.0 |
| | TIP-Adapter (2022) | 44.6 | 85.9 | 70.8 | 76.0 | 82.3 | 71.9 | 83.9 | 73.4 | 96.2 | 86.8 | 92.6 | 95.7 | 88.1 |
| | CLIP-Adapter (2024) | 34.2 | 71.4 | 59.4 | 74.2 | 74.0 | 62.6 | 80.2 | 69.8 | 92.9 | 87.1 | 92.3 | 94.9 | 86.2 |
| | PLOT++ (2022) | 46.7 | **92.0** | 71.4 | 76.0 | 84.6 | 74.1 | 85.3 | 72.6 | 97.6 | 87.1 | **93.6** | 96.0 | 88.7 |
| | KgCoOp (2023) | 36.5 | 76.2 | 68.7 | 73.3 | 74.8 | 65.9 | 81.7 | 70.4 | 93.4 | **87.2** | 93.2 | 95.2 | 86.9 |
| | ProGrad (2024) | 43.0 | 83.6 | 68.8 | 75.1 | 82.9 | 70.7 | 82.7 | 72.1 | 96.6 | 85.8 | 92.8 | 95.9 | 87.7 |
| | CLIP-LoRA (2024) | 54.7 | 90.7 | 73.0 | 76.0 | 86.0 | 76.1 | 86.2 | 73.4 | 97.9 | 84.2 | 91.6 | 96.1 | 88.2 |
| | +FMA (Ours) | **57.8** | 91.0 | **75.4** | **77.2** | **87.7** | **77.8**+1.7 | **87.1** | **73.5** | **99.1** | 85.1 | 91.6 | **96.5** | **88.8**+0.6 |

Table 3: **Model agnostic results**. We implemented FMA on several PEFT approaches using 16-shot settings.

| Method | Difficult | | | | | | Easy | | | | | | |
|---|---|---|---|---|---|---|---|---|---|---|---|---|---|
| | Aircraft | SAT | DTD | SUN | Cars | Avg | UCF | Net | Flowers | Food | Pets | Caltech | Avg |
| CLIP (2021) | 24.8 | 47.8 | 43.8 | 62.5 | 65.5 | 48.9 | 66.7 | 66.7 | 67.4 | 85.3 | 89.1 | 92.9 | 78.8 |
| +FMA | 38.1 | 84.3 | 69.5 | 73.5 | 78.6 | 68.9+20.0 | 82.0 | 70.4 | 97.5 | 87.0 | 92.8 | 95.9 | 87.6 +8.8 |
| CoOp (2022b) | 43.2 | 86.0 | 70.0 | 74.9 | 83.1 | 71.4 | 83.1 | 71.4 | 97.2 | 84.4 | 91.1 | 95.5 | 87.1 |
| +FMA | 47.6 | 88.1 | 73.1 | 75.9 | 85.4 | 74.0+2.6 | 84.4 | 72.5 | 98.2 | 85.0 | 91.4 | 95.7 | 87.9 +0.8 |
| CoCoOp (2022a) | 33.8 | 75.5 | 65.8 | 72.8 | 72.4 | 64.1 | 76.0 | 71.1 | 87.1 | 87.4 | 93.2 | 95.2 | 85.0 |
| +FMA | 36.9 | 86.9 | 71.9 | 73.4 | 73.5 | 68.5+4.4 | 80.3 | 71.9 | 94.5 | 87.8 | 93.4 | 95.6 | 87.3+2.3 |
| CLIP-Adapter (2024) | 33.8 | 70.4 | 59.3 | 74.3 | 74.2 | 62.4 | 80.1 | 71.6 | 93.6 | 87.1 | 92.4 | 94.9 | 86.6 |
| +FMA | 35.8 | 85.6 | 69.2 | 74.4 | 74.7 | 67.9+5.5 | 81.5 | 71.3 | 95.6 | 87.2 | 92.9 | 96.0 | 87.4+0.8 |
| CLIP-LoRA (2022) | 54.7 | 90.7 | 73.0 | 76.0 | 86.0 | 76.1 | 86.2 | 73.4 | 97.9 | 84.2 | 91.6 | 96.1 | 88.2 |
| +FMA | 57.8 | 91.0 | 75.4 | 77.2 | 87.7 | 77.8+1.7 | 87.1 | 73.5 | 99.1 | 85.1 | 91.6 | 96.5 | 88.8+0.6 |

not only reported the generalization (cross-dataset transfer) results, but also provided the adaptation (few-shot learning) performance and its harmonic mean.

**Results.** The results are shown in Table 1. While FMA improves the few-shot learning performance (first two columns), it doesn't result in further degradation of the generalization ability (middle two columns). Meanwhile, the improvement of the harmonic mean indicates FMA achieves a better trade-off between the adaptation and generalization ability. See more details in Appendix D.

## 5.3 ABLATION STUDY

**Architecture Agnostic.** In this section We evaluated whether our FMA is effective on other PEFT approaches. Specifically, we implemented FMA on pre-trained CLIP and four different PEFT methods: CoOp, CoCoOp, CLIP-Adapter, and CLIP-LoRA. For each PEFT approach, we first fine-tuned the pre-trained ViT-B/16 CLIP accordingly. After that, we extracted image and text features using the pre-trained or fine-tuned CLIP. Finally, we trained a velocity network on these features.

*Results.* As shown in Table 3, FMA achieves consistent performance improvements on all approaches, which demonstrates the architecture-agnostic ability of our framework. This means FMA can be easily adapted to different PEFT methods without significant performance degradation. Meanwhile, across all approaches, improvements on difficult datasets consistently surpass those on easy datasets.

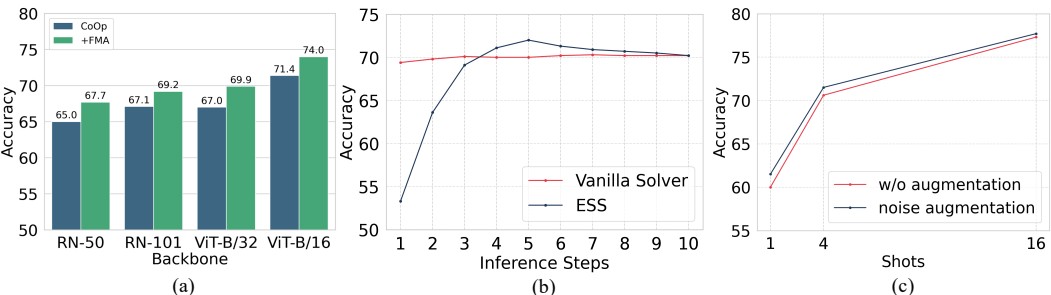

Figure 6: (a) Performance of different CLIP backbones on difficult datasets. (b) Ablation on different inference strategies. (c) Ablation on noise augmentation strategy. The average performance on 11 datasets is reported.

This further supports our conclusion that for difficult datasets with complex cross-modal distribution, multi-step rectification is more effective.

**Different CLIP Backbones.** This experiment was to evaluate whether FMA is effective on different CLIP backbones. We implemented FMA on top of CoOp and chose four different CLIP backbones: ResNet50, ResNet101, ViT-B/32, and ViT-B/16. Specifically, we first used a trained CoOp model to extract image and text features. After that, we trained a velocity network on these features according to our FMA framework. We reported the average performance on difficult datasets.

*Results.* The accuracy result is shown in Figure 6(a). Across all backbones, FMA achieves better performance than CoOp, demonstrating the effectiveness of our method. This also shows that FMA can be easily adapted to different backbones and further boost their performance.

**Different Inference Strategies.** To evaluate the influence of inference strategies on FMA, we conducted experiments with different numbers of inference steps using two kinds of strategies: Vanilla flow matching solver and our early stopping solver. Specifically, we used the default stepsize $h = 0.1$ and enumerated inference steps $M \in [0, 10]$. We conducted the experiments with a setting of 16 shots on the DTD dataset.

*Results.* As shown in Figure 6(b), as the number of inference steps increases, the performance of ESS will decrease after a certain step number (8 in this case). This is because training a perfect velocity field is extremely difficult, so more inference steps mean more accumulated errors. This suggests that choosing a proper number of inference steps is crucial for achieving good performance. Additionally, compared with the vanilla solver that is not sensitive to inference steps, ESS achieves better results when an appropriate inference step is chosen.

**Noise Augmentation.** To show the influence of noise augmentation in FMA, we trained two velocity networks on top of CLIP-LoRA, one with noise augmentation and another without any augmentation technique. We kept all other parts in FMA the same, such as the feature extraction and inference. In the inference, we use ESS to transform image features for classification.

*Results.* Performance on difficult datasets is shown in Figure 6(c). We can see that noise augmentation can improve the performance of FMA compared with no augmentation. And this improvement is consistent across different sizes of the training dataset, which proves the effectiveness of this strategy in the velocity field training process.

## 6 CONCLUSION

In this paper, we study how to achieve better cross-modality alignment based on pre-trained VLMs. Although many PEFT methods have been proposed, we find that these methods usually struggle on difficult datasets. In this paper, we conclude that these approaches are "one-step", which are not enough to decouple and align features in difficult datasets. To solve this, we propose our FMA framework to utilize the multi-step rectification ability of flow matching. In the training of FMA, we design two methods, coupling enforcement and noise augmentation, to learn a velocity field that can maintain class correspondence. Additionally, we propose an early-stopping solver for better inference performance. FMA is a plug-and-play module and we have conducted extensive experiments to demonstrate its effectiveness over a variety of benchmarks. We hope our research can explore a new potential opportunity, *i.e.*, multi-step rectification in the few-shot learning area.

## ACKNOWLEDGMENT

This work was supported by the National Natural Science Foundation of China Young Scholar Fund Category B (62522216), Hong Kong SAR RGC General Research Fund (16219025), National Natural Science Foundation of China Young Scholar Fund Category C (62402408), and Hong Kong SAR RGC Early Career Scheme (26208924).

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

## A    THEORETICAL GUARANTEE OF FMA

**Proposition 1** (Identity). *Consider an intermediate feature $x_t$, where $t \in [0,1]$, and $z_c \in p_1$ is its corresponding text feature. Then the marginal velocity field is identical to the condition velocity field of $x_t$:*

$$v_t(x_t) = v_t(x_t|z_c) \tag{7}$$

*Proof.* As we have discussed in the Coupling Enhancement strategy, the conditional probability $p_t(x_t|z)$ only assigns probability mass to features with the same label. The implicts that, if $x_t$ and $z_c$ belong to the same category, then for all $z' \notin z_c$, we have $p_t(x_t|z') = 0$. Then we have:

$$
\begin{aligned}
v_t(x_t) &= \int v_t(x_t|z)\frac{p_t(x_t|z)p_1(z)}{p_t(x_t)}\mathrm{d}z \\
&= v_t(x_t|z_c)\frac{p_t(x_t|z_c)p_1(z_c)}{p_t(x_t)} + \int_{z \neq z_c} v_t(x_t|z)\frac{p_t(x_t|z)p_1(z)}{p_t(x_t)}\mathrm{d}z \\
&= v_t(x_t|z_c)\frac{p_t(x_t|z_c)p_1(z_c)}{p_t(x_t)} + \int_{z \neq z_c} v_t(x_t|z)\frac{0 \cdot p_1(z)}{p_t(x_t)}\mathrm{d}z \\
&= v_t(x_t|z_c)\frac{p_t(x_t|z_c)p_1(z_c)}{p_t(x_t)} \\
&= v_t(x_t|z_c)\frac{p_t(x_t|z_c)p_1(z_c)}{\int p_t(x_t|z)p_t(z)\mathrm{d}z} \\
&= v_t(x_t|z_c)\frac{p_t(x_t|z_c)p_1(z_c)}{p_t(x_t|z_c)p_t(z_c) + \int_{z \neq z_c} p_t(x_t|z)p_t(z)\mathrm{d}z} \\
&= v_t(x_t|z_c)\frac{p_t(x_t|z_c)p_1(z_c)}{p_t(x_t|z_c)p_1(z_c) + \int_{z \neq z_c} 0 \cdot p_t(z)\mathrm{d}z} \\
&= v_t(x_t|z_c)\frac{p_t(x_t|z_c)p_1(z_c)}{p_t(x_t|z_c)p_1(z_c)} \\
&= v_t(x_t|z_c)
\end{aligned}
$$

**Proposition 2** (Classification). *Given any text embedding $z_c \in p_1$, if a conditional velocity field $v_t(x_t|z_c)$ is defined to generate $p_t(x_t|z_c)$, namely:*

$$x_0 \sim p_0, \quad \frac{\mathrm{d}x_t}{\mathrm{d}t} = v_t(x_t|z_c), \quad \Rightarrow x_t \sim p_t(x_t|z_c), \quad t \in (0,1] \tag{8}$$

*Then the marginal velocity $v_t(x_t)$, defined by*:

$$v_t(x_t) = \int v_t(x_t|z)\frac{p_t(x_t|z)p_1(z)}{p_t(x_t)}\mathrm{d}z \tag{9}$$

*can guarantee*:

$$x_1 = z_c \tag{10}$$

*where $x_1$ is estimated by:* $\mathrm{d}x_t = v_t(x_t)\mathrm{d}t$

*Proof.*

Consider the two transformation ODE using the marginal/conditional velocity:

$$x_0 \sim p_0, \quad \frac{\mathrm{d}x_t}{\mathrm{d}t} = v_t(x_t|z_c), \quad \Rightarrow x_t \sim p_t(x_t|z_c), \quad t \in (0,1] \tag{11}$$

$$x_0 \sim p_0, \quad \frac{\mathrm{d}x_t}{\mathrm{d}t} = v_t(x_t) \quad \Rightarrow x_t \sim p_t(x_t), \quad t \in (0,1] \tag{12}$$

Given the same intial image feature $x_0$, because $v_t(x_t) = v_t(x_t|z_c)$, these two ODEs are essentially the same one. By the Picard-Lindelöf theorem (assuming $v_t(x_t)$ and $v_t(x_t|z_c)$ is Lipschitz continuous,

which is standard in flow modeling), the solution to an ODE with a given initial condition is unique. Therefore, $x_1$ obatained by $v_t(x_t)$ also have:

$$x_1 \sim p_1(x_1|z_c) = \mathcal{N}(1 \cdot z_c + 0 \cdot x_0, 0) = z_c$$

By now, we have proved that the marginal velocity $v_t(x_t)$ can not only convert an image feature $x_0$ into a text feature $z \in p_1$, but also guarantee that this text feature is exactly of the same class as $x_0$, that is: $z = z_c$.

After that, we need to figure out how to learn $v_t(x_t)$ using a neural network $v_t^\theta(x_t)$. Intuitively, we can optimize it by solving a least square regression problem between $v_t(x_t)$ and $u_t^\theta(x_t)$:

$$\mathcal{L}_{FM}(\theta) = \mathbb{E}_{x_t \sim p_t, t \sim [0,1]}[\|u_t^\theta(x_t) - v_t(x_t)\|^2], \tag{13}$$

Because $v_t(x_t) = v_t(x_t|z)$, and $p_t(x_t|z) = \mathcal{N}(t \cdot z + (1-t) \cdot x_0, 0)$, we have $x_t = t \cdot z + (1-t) \cdot x_0$, then :

$$v_t(x_t) = v_t(x_t|z) = \frac{\mathrm{d}x_t}{\mathrm{d}t} = \frac{\mathrm{d}(t \cdot z + (1-t) \cdot x_0)}{\mathrm{d}t} = z - x_0$$

.

Therefore, the training objective becomes:

$$\mathcal{L}_{FM}(\theta) = \mathbb{E}_{z \sim p_1, x_0 \sim p_0(x0|z), t \sim [0,1]}[\|u_t^\theta(x_t) - (z - x_0)\|^2] \tag{14}$$

## B  ALGORITHM

Below is the training and inference algorithm of our FMA, respectively.

---

**Algorithm 1** Training

1: **Input:** paired image features and text features $\mathcal{D} = \{(x_0^i, z_c^i)\}_{i=0}^{N \times K}$
2: **repeat**
3:   $t \sim \mathcal{U}([0,1]), (x_0, z_c) \sim \mathcal{D}$
4:   $x_t = (1-t)x_0 + tz_c$
5:   $\hat{x}_t \sim \mathcal{N}(\hat{x}_t|x_t, t \cdot (1-t) \cdot \sigma^2(x_t)))$
6:   Take gradient descent on $u_t^\theta(\hat{x}_t)$
7: **until** converges

**Algorithm 2** Inference

1: **Input:** image features $x_0$, trained $u_t^\theta(x_t)$, steps $M$, stepsize $h$
2: $t = 0$
3: **for** $n = 1$ to $M$ **do**
4:   $x_{t+h} = x_t + h \cdot u_t^\theta(x_t)$
5:   $t = t + h$
6: **end for**
7: **return** $x_{h \cdot M}$ as new image feature

---

## C  MODEL AGNOSTIC EXPERIMENT

This section provides more detail of Table 3: standard deviation(Table 4), statistical significance(Table 5), and corresponding inference steps(Table 6).

Table 4: **Details on standard deviations on Table 3**. The bottom right gray number next to each FMA's result is the corresponding standard deviation.

| Method | Difficult | | | | | | Easy | | | | | | |
|---|---|---|---|---|---|---|---|---|---|---|---|---|---|
| | Aircraft | SAT | DTD | SUN | Cars | Avg | UCF | Net | Flowers | Food | Pets | Caltech | Avg |
| CLIP | 24.8 | 47.8 | 43.8 | 62.5 | 65.5 | 48.9 | 66.7 | 66.7 | 67.4 | 85.3 | 89.1 | 92.9 | 78.0 |
| +FMA | $38.1_{\pm 1.0}$ | $84.3_{\pm 1.2}$ | $69.5_{\pm 0.4}$ | $73.5_{\pm 0.8}$ | $78.6_{\pm 0.8}$ | $68.8_{\pm 1.3}$ | $82.0_{\pm 1.1}$ | $70.4_{\pm 0.3}$ | $97.5_{\pm 2.0}$ | $87.0_{\pm 0.2}$ | $92.8_{\pm 0.4}$ | $95.9_{\pm 0.2}$ | $87.6_{\pm 0.3}$ |
| CoOp | 43.2 | 86.0 | 70.0 | 74.9 | 83.1 | 71.4 | 83.1 | 71.4 | 97.2 | 84.4 | 91.1 | 95.5 | 87.1 |
| +FMA | $47.6_{\pm 0.7}$ | $88.1_{\pm 0.3}$ | $73.1_{\pm 0.6}$ | $75.9_{\pm 0.2}$ | $85.4_{\pm 0.4}$ | $74.0_{\pm 0.4}$ | $84.4_{\pm 0.2}$ | $72.5_{\pm 0.2}$ | $98.2_{\pm 0.0}$ | $85.0_{\pm 0.1}$ | $91.4_{\pm 0.3}$ | $95.7_{\pm 0.1}$ | $87.9_{\pm 0.2}$ |
| CoCoOp | 33.8 | 75.5 | 65.8 | 72.8 | 72.4 | 64.1 | 76.0 | 71.1 | 87.1 | 87.4 | 93.2 | 95.2 | 85.0 |
| +FMA | $36.9_{\pm 0.6}$ | $86.9_{\pm 1.1}$ | $71.9_{\pm 0.8}$ | $73.4_{\pm 0.8}$ | $73.5_{\pm 0.4}$ | $68.5_{\pm 0.9}$ | $80.3_{\pm 1.3}$ | $71.9_{\pm 0.1}$ | $94.5_{\pm 0.5}$ | $87.8_{\pm 0.2}$ | $93.4_{\pm 0.2}$ | $95.6_{\pm 0.2}$ | $87.3_{\pm 0.4}$ |
| CLIP-Adapter | 33.8 | 70.4 | 59.3 | 74.3 | 74.2 | 62.4 | 80.1 | 71.6 | 93.6 | 87.1 | 92.4 | 94.9 | 86.6 |
| +FMA | $35.8_{\pm 0.3}$ | $85.6_{\pm 1.2}$ | $69.2_{\pm 0.1}$ | $74.4_{\pm 0.1}$ | $74.7_{\pm 0.1}$ | $67.9_{\pm 0.7}$ | $81.5_{\pm 0.3}$ | $71.3_{\pm 0.2}$ | $95.6_{\pm 0.4}$ | $87.2_{\pm 0.2}$ | $92.9_{\pm 0.0}$ | $96.0_{\pm 0.3}$ | $87.4_{\pm 0.2}$ |
| CLIP-LoRA | 54.7 | 90.7 | 73.0 | 76.0 | 86.0 | 76.1 | 86.2 | 73.4 | 97.9 | 84.2 | 91.6 | 96.1 | 88.2 |
| +FMA | $57.8_{\pm 0.7}$ | $91.0_{\pm 0.0}$ | $75.4_{\pm 0.5}$ | $77.2_{\pm 0.2}$ | $87.7_{\pm 0.3}$ | $77.8_{\pm 0.3}$ | $87.1_{\pm 0.1}$ | $73.5_{\pm 0.1}$ | $99.1_{\pm 0.4}$ | $85.1_{\pm 0.2}$ | $91.6_{\pm 0.1}$ | $96.5_{\pm 0.1}$ | $88.8_{\pm 0.2}$ |

Table 5: **Details on p-value on Table 3**. The baseline indicates which method we implement for FMA.

| Baseline | Difficult | | | | | | Easy | | | | | | |
|---|---|---|---|---|---|---|---|---|---|---|---|---|---|
| | Aircraft | SAT | DTD | SUN | Cars | Avg | UCF | Net | Flowers | Food | Pets | Caltech | Avg |
| CLIP | 0.000 | 0.000 | 0.000 | 0.000 | 0.001 | 0.000 | 0.000 | 0.000 | 0.000 | 0.001 | 0.000 | 0.000 | 0.000 |
| CoOp | 0.008 | 0.006 | 0.015 | 0.142 | 0.021 | 0.004 | 0.032 | 0.185 | 0.009 | 0.254 | 0.347 | 0.289 | 0.035 |
| CoCoOp | 0.048 | 0.000 | 0.002 | 0.375 | 0.210 | 0.001 | 0.001 | 0.245 | 0.000 | 0.322 | 0.415 | 0.076 | 0.008 |
| CLIP-Adapter | 0.015 | 0.000 | 0.000 | 0.835 | 0.475 | 0.000 | 0.028 | 0.612 | 0.003 | 0.802 | 0.124 | 0.004 | 0.045 |
| CLIP-LoRA | 0.008 | 0.374 | 0.024 | 0.046 | 0.018 | 0.012 | 0.034 | 0.876 | 0.001 | 0.095 | 0.871 | 0.089 | 0.041 |

Table 6: **Details on inference steps ($M$) for different baselines corresponding to the results in Table 3.** The average steps are rounded to one decimal place.

| Baseline | FGVC | EuroSAT | DTD | SUN | Cars | UCF | Net | Flow | Food | Pets | Cal | Avg |
|---|---|---|---|---|---|---|---|---|---|---|---|---|
| CLIP | 7 | 6 | 5 | 3 | 6 | 4 | 1 | 5 | 1 | 1 | 2 | 3.7 |
| CoOp | 2 | 2 | 2 | 3 | 1 | 4 | 1 | 1 | 1 | 1 | 1 | 1.7 |
| CoCoOp | 7 | 2 | 6 | 6 | 4 | 9 | 2 | 3 | 2 | 3 | 3 | 4.3 |
| CLIP-Adapter | 3 | 7 | 5 | 3 | 2 | 2 | 1 | 5 | 1 | 2 | 3 | 3.1 |
| CLIP-LoRA | 3 | 5 | 2 | 3 | 3 | 2 | 2 | 6 | 2 | 1 | 2 | 2.8 |

## D    GENERALIZATION EXPERIMENT

This section provides the detail of Table 1. For each dataset, we report its generalization performance on five baselines and FMA in Table 7.

Table 7: **Cross-dataset transfer evaluation.** We train models on ImageNet (Source) and evaluate their generalization ability on 10 other unseen datasets (Target).

| Method | Source | Target Datasets | | | | | | | | | |
|---|---|---|---|---|---|---|---|---|---|---|---|
| | ImageNet | FGVC | EuroSAT | DTD | SUN | Cars | UCF | Flow | Food | Pets | Cal |
| CoOp | 71.4 | 16.2 | 49.3 | 35.7 | 58.0 | 63.2 | 60.9 | 59.7 | 80.8 | 86.9 | 91.1 |
| +FMA | 72.5 | 16.1 | 49.8 | 36.5 | 57.6 | 62.4 | 61.6 | 58.7 | 80.2 | 86.5 | 91.5 |
| Adapter | 71.6 | 23.2 | 44.0 | 42.6 | 63.6 | 61.9 | 64.4 | 68.9 | 84.2 | 86.8 | 94.9 |
| +FMA | 71.3 | 23.0 | 45.1 | 45.4 | 63.9 | 61.1 | 64.2 | 68.7 | 83.3 | 88.2 | 95.9 |
| CLIP | 66.7 | 24.8 | 48.3 | 44.1 | 62.6 | 65.5 | 67.5 | 70.7 | 85.9 | 89.0 | 93.3 |
| +FMA | 70.4 | 24.0 | 49.1 | 46.6 | 63.1 | 64.2 | 66.8 | 68.9 | 85.2 | 88.5 | 95.7 |
| LoRA | 73.4 | 22.5 | 32.5 | 44.2 | 66.5 | 63.2 | 66.0 | 70.0 | 84.5 | 89.1 | 92.9 |
| +FMA | 73.5 | 22.6 | 37.4 | 44.4 | 66.3 | 63.0 | 67.3 | 69.3 | 83.6 | 89.1 | 93.6 |
| CoCoOp | 71.1 | 23.7 | 45.7 | 46.0 | 66.5 | 65.8 | 69.4 | 70.2 | 86.0 | 90.3 | 94.8 |
| +FMA | 71.9 | 24.3 | 49.5 | 47.3 | 66.9 | 65.0 | 70.0 | 70.0 | 85.2 | 90.5 | 95.5 |

