| $46.4_{\pm1.1}$ | $87.9_{\pm1.4}$ | $72.0_{\pm0.3}$ | $73.3_{\pm0.8}$ | $83.1_{\pm1.0}$ | $72.5_{\pm1.4}$ | $82.9_{\pm0.8}$ | $71.0_{\pm0.5}$ | $98.5_{\pm2.1}$ | $87.0_{\pm0.2}$ | $92.7_{\pm0.3}$ | $95.8_{\pm0.2}$ | $88.0_{\pm0.4}$ |
| CoOp (?) | 43.2 | 86.0 | 70.0 | 74.9 | 83.1 | 71.4 | 83.1 | 71.4 | 97.2 | 84.4 | 91.1 | 95.5 | 87.1 |
| +FMA | $47.6_{\pm0.7}$ | $88.1_{\pm0.3}$ | $73.1_{\pm0.6}$ | $75.9_{\pm0.2}$ | $84.4_{\pm0.2}$ | $73.8_{\pm0.4}$ | $85.4_{\pm0.3}$ | $72.5_{\pm0.2}$ | $98.2_{\pm0.0}$ | $85.0_{\pm0.1}$ | $91.4_{\pm0.3}$ | $95.7_{\pm0.1}$ | $88.0_{\pm0.2}$ |
| CoCoOp (?) | 33.8 | 75.5 | 65.8 | 72.8 | 76.0 | 64.8 | 72.4 | 71.1 | 87.1 | 87.4 | 93.2 | 95.2 | 84.4 |
| +FMA | $36.9_{\pm0.6}$ | $86.9_{\pm1.1}$ | $71.9_{\pm0.8}$ | $73.4_{\pm0.8}$ | $80.3_{\pm1.3}$ | $69.9_{\pm0.9}$ | $73.5_{\pm0.3}$ | $71.9_{\pm0.1}$ | $94.5_{\pm0.5}$ | $87.8_{\pm0.2}$ | $93.4_{\pm0.2}$ | $95.6_{\pm0.2}$ | $86.1_{\pm0.4}$ |
| CLIP-Adapter (?) | 33.8 | 70.4 | 59.3 | 74.3 | 80.1 | 63.6 | 74.2 | 71.6 | 93.6 | 87.1 | 92.4 | 94.9 | 85.6 |
| +FMA | $35.8_{\pm0.3}$ | $85.6_{\pm1.2}$ | $69.2_{\pm0.1}$ | $74.4_{\pm0.1}$ | $81.5_{\pm0.3}$ | $69.3_{\pm0.7}$ | $74.7_{\pm0.1}$ | $71.3_{\pm0.2}$ | $95.6_{\pm0.4}$ | $87.2_{\pm0.2}$ | $92.9_{\pm0.0}$ | $96.0_{\pm0.1}$ | $86.3_{\pm0.2}$ |
| CLIP-LoRA (?) | 54.7 | 90.7 | 73.0 | 76.0 | 86.2 | 76.1 | 86.0 | 73.4 | 97.9 | 84.2 | 91.6 | 96.1 | 88.2 |
| +FMA | $57.8_{\pm0.7}$ | $91.0_{\pm0.0}$ | $75.4_{\pm0.5}$ | $77.2_{\pm0.2}$ | $87.1_{\pm0.1}$ | $77.7_{\pm0.3}$ | $87.7_{\pm0.3}$ | $73.5_{\pm0.1}$ | $99.1_{\pm0.4}$ | $85.1_{\pm0.2}$ | $91.6_{\pm0.1}$ | $96.5_{\pm0.1}$ | $88.9_{\pm0.2}$ |

Table 2: **Details on p-value on Table ??**. The baseline indicates which method we implement for FMA.

| Baseline | Difficult | | | | | | Easy | | | | | | |
|---|---|---|---|---|---|---|---|---|---|---|---|---|---|
| | Aircraft | SAT | DTD | SUN | UCF | Avg | Cars | Net | Flowers | Food | Pets | Caltech | Avg |
| CLIP (?) | 0.000 | 0.000 | 0.000 | 0.000 | 0.000 | 0.000 | 0.001 | 0.000 | 0.000 | 0.001 | 0.000 | 0.000 | 0.000 |
| CoOp (?) | 0.008 | 0.006 | 0.015 | 0.142 | 0.032 | 0.004 | 0.021 | 0.185 | 0.009 | 0.254 | 0.347 | 0.289 | 0.035 |
| CoCoOp (?) | 0.048 | 0.000 | 0.002 | 0.375 | 0.001 | 0.001 | 0.210 | 0.245 | 0.000 | 0.322 | 0.415 | 0.076 | 0.008 |
| CLIP-Adapter (?) | 0.065 | 0.000 | 0.000 | 0.835 | 0.028 | 0.000 | 0.475 | 0.612 | 0.003 | 0.802 | 0.124 | 0.004 | 0.045 |
| CLIP-LoRA (?) | 0.008 | 0.374 | 0.024 | 0.046 | 0.034 | 0.012 | 0.018 | 0.876 | 0.001 | 0.095 | 0.871 | 0.089 | 0.041 |

Table 3: **Details on inference steps ($M$) for different baselines corresponding to the results in Table ??.** The average steps are rounded to one decimal place.

| Baseline | FGVC | EuroSAT | DTD | SUN | UCF | Cars | Net | Flow | Food | Pets | Cal | **Avg** |
|---|---|---|---|---|---|---|---|---|---|---|---|---|
| CLIP | 7 | 6 | 5 | 3 | 4 | 6 | 1 | 5 | 1 | 1 | 2 | 3.7 |
| CoOp | 2 | 2 | 2 | 3 | 4 | 1 | 1 | 1 | 1 | 1 | 1 | 1.7 |
| CoCoOp | 7 | 2 | 6 | 6 | 9 | 4 | 2 | 3 | 2 | 3 | 3 | 4.3 |
| CLIP-Adapter | 3 | 7 | 5 | 3 | 2 | 2 | 1 | 5 | 1 | 2 | 3 | 3.1 |
| CLIP-LoRA | 3 | 5 | 2 | 3 | 2 | 3 | 2 | 6 | 2 | 1 | 2 | 2.8 |

## D  MORE ABLATION STUDY.

**Adaptive inference strategy.** We implemented an adaptive strategy based on the logit margin. During the inference flow process at timestep $t$, we calculate the logits $L_t$. The inference is terminated early if the margin between the top-1 and top-2 class predictions exceeds a pre-defined threshold $\epsilon$:

$$Top1(L_t) - Top2(L_t) > \epsilon$$

We evaluated this strategy on the "Difficult" dataset group using the ViT-B/16 backbone under the 16-shot setting. We tested three different threshold values: $\epsilon \in \{0.001, 0.0007, 0.0005\}$. See Table B.

**Noise Augmentation Strategy.**The noise is injected as $x_t \sim \mathcal{N}(x_t, \Sigma_t)$, where $\Sigma_t = k \cdot s(t)$. For Schedule $s(t)$: We adopted the symmetric schedule $s(t) = t(1-t)$, inspired by the Diffusion Schrodinger Bridge. See Table B.

**Different ODE Solvers.** To investigate whether higher-order solvers (which offer lower truncation errors for curved trajectories) provide better alignment accuracy than the basic Euler method, we evaluated three common numerical solvers under the same budget of inference steps ($M$):Euler Method, Heun's Method, and Runge-Kutta (RK4). See Table B.

Table 4: **Ablation study on adaptive inference strategies.** We compare our proposed Early Stopping Solver (ESS) with fixed steps against an adaptive strategy based on logit margin thresholds ($\epsilon$). The best results are highlighted in bold.

| Method | Aircraft | SAT | DTD | SUN | UCF | Avg |
|---|---|---|---|---|---|---|
| CLIP (Zero-shot) | 24.8 | 47.8 | 43.8 | 62.5 | 66.7 | 49.1 |
| **ESS (Ours, Fixed $M$)** | **46.4** | **87.9** | **72.0** | **73.3** | **83.1** | **72.5** |
| Adaptive ($\epsilon = 0.0010$) | 42.1 | 86.9 | **72.0** | 71.9 | **83.1** | 71.2 |
| Adaptive ($\epsilon = 0.0007$) | 46.0 | 86.1 | 71.3 | 72.4 | 82.7 | 71.7 |
| Adaptive ($\epsilon = 0.0005$) | **46.4** | **87.9** | 70.2 | **73.3** | 81.5 | 71.8 |

Table 5: **Ablation study on noise magnitude.** We evaluate the impact of different noise scaling factors $k$ in the noise augmentation term $\hat{x}_t \sim \mathcal{N}(x_t, k \cdot t(1 - t))$.

| Method | Aircraft | SAT | DTD | SUN | UCF | Avg |
|---|---|---|---|---|---|---|
| CLIP (Zero-shot) | 24.8 | 47.8 | 43.8 | 62.5 | 66.7 | 49.1 |
| **FMA ($k = \sigma^2(x_t)$)** | **46.4** | **87.9** | **72.0** | 73.3 | **83.1** | **72.5** |
| FMA ($k = 2\sigma^2(x_t)$) | 45.8 | 87.2 | 71.1 | **83.2** | 71.2 | 71.7 |
| FMA ($k = 5\sigma^2(x_t)$) | 43.6 | 83.8 | 69.1 | 79.5 | 68.8 | 69.0 |
| FMA ($k = 1$) | 37.9 | 78.1 | 64.2 | 74.2 | 54.1 | 61.7 |

Table 6: **Ablation study on ODE solvers during inference.** We compare the standard Euler method (1st order) against higher-order solvers (Heun, RK4).

| Method | Aircraft | SAT | DTD | SUN | UCF | Avg |
|---|---|---|---|---|---|---|
| CLIP (Zero-shot) | 24.8 | 47.8 | 43.8 | 62.5 | 66.7 | 49.1 |
| **FMA (Euler)** | **46.4** | **87.9** | **72.0** | 73.3 | **83.1** | **72.5** |
| FMA (Heun) | **46.4** | 87.7 | 71.8 | **73.4** | **83.1** | **72.5** |
| FMA (RK4) | **46.4** | 87.7 | 71.9 | 73.3 | **83.1** | **72.5** |

# E  GENERALIZATION EXPERIMENT.

This section provides the detail of Table **??**. For each dataset, we report its generalization performance on five baselines and FMA in Table B.

Table 7: **Cross-dataset transfer evaluation.** We train models on ImageNet (Source) and evaluate their generalization ability on 10 other unseen datasets (Target).

| Method | Source | Target Datasets | | | | | | | | | |
|---|---|---|---|---|---|---|---|---|---|---|---|
| | ImageNet | FGVC | EuroSAT | DTD | SUN | UCF | Cars | Flow | Food | Pets | Cal |
| CoOp | 71.4 | 16.2 | 49.3 | 35.7 | 58.0 | 60.9 | 63.2 | 59.7 | 80.8 | 86.9 | 91.1 |
| +FMA | 72.4 | 16.1 | 49.8 | 36.5 | 57.6 | 61.6 | 62.4 | 58.7 | 80.2 | 86.5 | 91.5 |
| Adapter | 71.6 | 23.2 | 44.0 | 42.6 | 63.6 | 64.4 | 61.9 | 68.9 | 84.2 | 86.8 | 94.9 |
| +FMA | 71.3 | 23.0 | 45.1 | 45.4 | 63.9 | 64.2 | 61.1 | 68.7 | 83.3 | 88.2 | 95.9 |
| CLIP | 66.7 | 24.8 | 48.3 | 44.1 | 62.6 | 67.5 | 65.5 | 70.7 | 85.9 | 89.0 | 93.3 |
| +FMA | 71.0 | 24.0 | 49.1 | 46.6 | 63.1 | 66.8 | 64.2 | 68.9 | 85.2 | 88.5 | 95.7 |
| LoRA | 73.4 | 22.5 | 32.5 | 44.2 | 66.5 | 66.0 | 63.2 | 70.0 | 84.5 | 89.1 | 92.9 |
| +FMA | 73.5 | 22.6 | 37.4 | 44.4 | 66.3 | 67.3 | 63.0 | 69.3 | 83.6 | 89.1 | 93.6 |
| CoCoOp | 71.1 | 23.7 | 45.7 | 46.0 | 66.5 | 69.4 | 65.8 | 70.2 | 86.0 | 90.3 | 94.8 |
| +FMA | 71.9 | 24.3 | 49.5 | 47.3 | 66.9 | 70.0 | 65.0 | 70.0 | 85.2 | 90.5 | 95.5 |

# F COMPUTATIONAL COST.

We conducted a comprehensive comparison of computational costs between FMA and five baseline methods. For a fair comparison, we fixed the training/test batch size at 32 and measured the average training time per iteration and peak memory usage on a single NVIDIA RTX 3090 GPU. See Table B.

Table 8: Comparison of computational cost (Training Time, Memory, and Inference Latency).

| Method | Training Time | Training Memory | Inference Time/CLIP Inference Time |
|---|---|---|---|
| CLIP | - | - | 1 (10.2ms) |
| CoOp | 712ms | 2770MB | 1.2 |
| CoCoOp | 1912ms | 12318MB | 3.2 |
| CLIP-Adapter | 157ms | 1376MB | 1.0 |
| CLIP-LoRA | 331ms | 5406MB | 1.3 |
| FMA | 198ms | 2842MB | 1.2 ($M$=1) |