# OpenReview forum: "Exploring Cross-Modal Flows for Few-Shot  Learning"
_ICLR.cc/2026/Conference — ICLR 2026 Poster_

### Official Review · Reviewer_girA · 2025-10-23

**Soundness:** 3
**Presentation:** 3
**Contribution:** 3
**Rating:** 6
**Confidence:** 3

**Summary:**

This paper introduces a new framework called Flow Matching Alignment (FMA) to improve feature alignment between visual and textual modalities in few-shot learning. The authors observe that current parameter-efficient fine-tuning (PEFT) methods—such as prompt tuning, adapter tuning, and LoRA—perform only one-step feature adjustments, which are insufficient for complex datasets where image–text features are highly entangled. FMA leverages the multi-step rectification ability of flow matching by learning a cross-modal velocity field that iteratively transforms image features toward corresponding text embeddings, achieving finer alignment. To ensure stability and correctness, the method incorporates three key designs: coupling enforcement (to maintain class correspondence), noise augmentation (to mitigate data scarcity), and an early-stopping solver (to prevent over-transformation during inference). Experiments across 11 benchmarks and multiple backbones show that FMA consistently outperforms existing PEFT methods.

**Strengths:**

1. FMA introduces flow matching to few-shot learning. By formulating traditional PEFT methods as one-step updates, FMA enables more precise iterative alignment between visual and textual features. As argued by athe uthors, FMA better handles entangled multimodal distributions, especially in challenging datasets.
2. The framework is architecture-independent and can be integrated with various pre-trained vision-language models (e.g., CLIP, CoOp, LoRA) without altering their internal structures.

**Weaknesses:**

1. The multi-step flow matching process requires iterative training and inference, which increases computational cost compared to traditional one-step PEFT methods, potentially limiting scalability for large datasets or real-time applications.
2. The method relies on carefully chosen parameters such as the number of inference steps, step size, and noise schedule. Suboptimal tuning can lead to degraded performance or instability during alignment. Especially when flow matching originates from generative modeling, and its adaptation to supervised classification tasks lacks rigorous theoretical grounding, in terms of convergence and optimal stopping criteria.

**Questions:**

1. As discussed in the weakness part, is there any theoretical guarantee of convergence and stability of FMA? Given that flow matching originates from generative modeling, what are the theoretical conditions under which FMA ensures convergence to the correct class-aligned distribution in supervised learning settings?

---

> ### Author Response · Authors · 2025-11-26
> **Response to Reviewer girA (1/1)**
>
> We sincerely appreciate your constructive comments. We detail our response below and have corrected the corresponding part in our revision.
>
> > Q1: As discussed in the weakness part, is there any theoretical guarantee of convergence and stability of FMA? Given that flow matching originates from generative modeling, what are the theoretical conditions under which FMA ensures convergence to the correct class-aligned distribution in supervised learning settings?
>
> For a formal guarantee, we have added a section in the revised manuscript to provide a theoretical background for FMA. Generally, the theory can be summarized as follows: (For more details, see Section 4 and Appendix A.)
> - The target of FMA is to find a marginal velocity $v_t(x_t)$, which can meet the following two requirements: 1) convert image features into text features; **2) make sure the transferred text features are the correct categories of the starting image features.** Because canonical flow matching cannot guarantee 2), we need to design a new framework, namely FMA.
> - We proved that the Coupling Enhancement is equivalent to defining the conditional probability path $p_t(x_t|z_c)$, when $t=0$.
> - Similar with canonical flow matching, we define $p_t(x_t|z_c)$ as linear interpolation when $t \neq 0$, then prove under a certain definition, $v_t(x_t)$can meet requirement 1.
> - Under Coupling Enhancement, we prove that $v_t(x_t)= v_t(x_t|z_c)$ in theory, where $z_c$ is the corresponding text embedding of $x_t$.
> - Using $v_t(x_t)= v_t(x_t|z_c)$, we prove $v_t(x_t)$ will meet requirement 2, which means it can guarantee that the transferred feature belongs to the same category as the initial image feature.
>
> For more details, see Section 4 and Appendix A.
>
> > W1: The multi-step flow matching process requires iterative training and inference, which increases computational cost compared to traditional one-step PEFT methods, potentially limiting scalability for large datasets or real-time applications.
>
> We conducted a comprehensive comparison of computational costs between FMA and five baseline methods.
>
> **Experimental Setup:** For a fair comparison, we fixed the training/test batch size at 32 and measured the average training time per iteration and peak memory usage on a single NVIDIA RTX 3090 GPU. The results are summarized below:
>
> |Method|Training Time |Training Memory|Inference time/(CLIP inference time)|
> |-|-|-|-|
> |CLIP|-|-|1 (10.2ms)|
> |CoOp|712ms|2770MB|1.2|
> |CoCoOp|1912ms|12318MB|3.2|
> |CLIP-Adapter|157ms|1376MB|1.0|
> |CLIP-LoRA|331ms|5406MB|1.3
> |FMA|198ms |2842MB|1.2 ($M$=1)
>
>
> **Analysis & Conclusions:**
>
> 1.  **High Training Efficiency:** As shown in the table, FMA is **significantly faster** than prompt-tuning methods (e.g., ~3.5x faster than CoOp) and comparable to the most efficient Adapter-based methods. This efficiency stems from our flow matching objective, which does not require backpropagation through the heavy text encoder during training.
>
> 2.  **Flexible Inference Trade-off:** We acknowledge that multi-step inference (M>1) increases latency. However, FMA offers a flexible **trade-off between speed and accuracy**.  For example, the performance of CLIP+FMA on EuroSAT, varying different $M$, is as follows.  **M=1** (which has a similar inference cost to CoOp) already achieves a massive improvement over Zero-Shot (47.8->73.6):
>
> |Steps|0(Zero-Shot)|1|2|3|4|5|
> |-|-|-|-|-|-|-|
> |Acc|47.8|73.6|83.1|85.8|87.0|87.5|
>
>
> 3.  **Contextualizing Efficiency:** In the few-shot learning community, complex models like CoCoOp (which is ~3x slower in inference) are widely accepted because they solve critical issues like generalization. Similarly, FMA tackles the difficult "entangled distribution" problem. Given its superior training speed and flexible inference steps, we believe FMA presents a highly competitive solution.
>
>
>
> > W2: When flow matching originates from generative modeling, its adaptation to supervised classification tasks lacks rigorous theoretical grounding.
>
> Thanks for the concern.  Theoretical guarantee is given in the reply to Q1.

---

> > ### Comment · Reviewer_girA · 2025-11-26
> >
> > Hi,
> >
> > I would like to thank the authors for the thorough rebuttal. Most of my concerns have been addressed and I would like to maintain my current rating.

---

### Official Review · Reviewer_CrjH · 2025-10-27

**Soundness:** 3
**Presentation:** 3
**Contribution:** 3
**Rating:** 6
**Confidence:** 4

**Summary:**

The paper reframes PEFT as a one-step adjustment problem that fails on difficult datasets and proposes Flow Matching Alignment (FMA), which learns a velocity field to iteratively transport image features toward their ground-truth text features. It introduces coupling enforcement to preserve class correspondence, noise augmentation to combat data sparsity and manifold collapse, and an early-stopping solver (ESS) that classifies from intermediate states to avoid late-stage drift. FMA is plug-and-play across CLIP and multiple PEFT backbones and shows consistent gains on 11 benchmarks, especially on difficult datasets, with ablations supporting each component.

**Strengths:**

The diagnosis of one-step PEFT limitations is convincing; the method is simple, modular, and effective across backbones; experiments are comprehensive; and the early-stopping insight is well-supported by empirical phenomena.

**Weaknesses:**

Lack of formal guarantees for coupling assumptions, reliance on validation to set inference steps, missing comparisons with higher-order ODE solvers, coarse difficulty metric, and limited analysis of compute trade-offs and failure modes.

**Questions:**

1. Can ESS be made adaptive without validation tuning (e.g., stop on sufficient logit margin, small velocity norm, or diminishing logit gains)?
2. How is σ(xt) chosen in practice, and how sensitive are results to the noise magnitude/schedule? Any benefit from uncertainty- or density-adaptive noise?
3. Did you try higher-order or adaptive ODE solvers (Heun/RK) to reduce truncation error and improve margins at the same step budget?
4. Is fixed pairing too rigid for multi-modal classes? Would transporting toward multiple positive prototypes or a class subspace help?
5. Have you considered adding a discriminative loss on intermediate states (e.g., contrastive/margin) to align transport with classification throughout the trajectory?
6. Can you report detailed overhead (velocity net size, train/infer time, average ESS steps) and scaling with number of classes?

---

> ### Author Response · Authors · 2025-11-26
> **Response to Reviewer CrjH(1/3)**
>
> We thank the reviewer for their positive assessment of our work, and we address the related questions below.
>
> >Q1:  Can ESS be made adaptive without validation tuning (e.g., stop on sufficient logit margin, small velocity norm, or diminishing logit gains)?
>
> In this paper,  we rely on a validation set to determine the inference steps M strictly to **follow the standard evaluation protocol in Few-Shot Learning** (similar to how Tip-Adapter/CLIP-Adapter tunes $\alpha$  on validation sets). Besides, we agree that adaptive stopping has the potential to obtain higher. However, most adaptive strategies still rely on some hyperparameters. Let's take the three strategies you mentioned, for example.
> 1. Stop on a sufficient logit margin. Given an image feature, first calculate the logits $L_t$ at timestep $t$, if $\mathrm{Top1}(L_t) - \mathrm{Top2}(L_t) > \epsilon$, then we can conclude that the velocity is confident enough to give a decision, thus stop the inference at the current timestep.
>  However, it is difficult to determine the value of $\epsilon$, and for different values of $\epsilon$, related results are as follows (16-shot, ViT-B/16, CLIP+FMA). For different datasets, we need different $\epsilon$ to obtain better results.  In this situation, $\epsilon$  is another hyperparameter, just like $M$, and we still need validation tuning for this strategy.
>
> |Method | Aircraft | SAT | DTD | SUN | UCF | Avg|
> |-|-|-|-|-|-|-|
> |CLIP|24.8|47.8|43.8|62.5|66.7|49.1|
> |ESS|**46.4**|**87.9**|**72.0**|**73.3**|**83.1**|**72.5**
> |$\epsilon$=0.001|42.1|86.9|**72.0**|71.9|**83.1**|71.2|
> |$\epsilon$=0.0007|46.0|86.1|71.3|72.4|82.7|71.7|
> |$\epsilon$=0.0005|**46.4**|**87.9**|70.2|**73.3**|81.5|71.8|
>
>
>
> 2. Small velocity norm.  We would like to clarify that since FMA employs the Rectified Flow objective (Eq. 1),  the model learns to approximate a straight trajectory with constant velocity ($v_t = x_1 - x_0$​). Therefore, **the velocity norm tends to remain stable rather than decaying to zero**, making it a less reliable stopping signal in our specific framework.
>
> 3. Diminishing logit gains. This strategy means:  if the improvement of the largest logit is less than a margin:$\mathrm{Top1}(L_{t+1}) - \mathrm{Top1}(L_t) < \delta$, then stop the inference.  This is similar to the "Stop on sufficient logit margin" strategy, which means we still need a validation set.
>
> So why usually a validation tuning is necessary? If we can train a perfect velocity, then the flow toward $t=1$ will always work.  However, due to the limited data and highly entangled feature space, the trained velocity is usually imperfect. Meanwhile,  there is still no tool to analyze how imperfect a velocity is in the flow matching theory. Therefore, tuning on a validation set is the most practical method.
>
>
> > Q2:  How is σ(xt) chosen in practice, and how sensitive are results to the noise magnitude/schedule? Any benefit from uncertainty- or density-adaptive noise?
>
>
> We apologize for the insufficient details.
>
> 1. $\sigma^2(x_t)$ in $\hat{x_t} \sim\mathcal{N}(\hat{x_t}|x_t,t\cdot(1-t)\cdot\sigma^2(x_t))$. Here $\sigma^2(x_t)\in\mathbb{R}$  is the standard deviation of $x_t\in \mathbb{R}^d$: $\sigma^2(x_t)=\sqrt{\sum_{i=1}^d(x_t^i-\mu(x_t))^2} /d$, where $\mu(x_t) =\sum_{i=1}^dx^i_t/d$ is the mean of $x_t$. $x^i_t\in \mathbb{R}$ is the i-th dimension of $x_t$. This term makes sure the noise strength is constrained by the statistical characteristic of the feature itself.
>
> 2. Noise magnitude/schedule. We found that compared with the noise schedule, the noise magnitude is of more significance.  First the schedule in the form of $t\cdot(1-t)$ is inspired by **Diffusion Schrodinger Bridge**, which is widely used in tasks like image2image translation. Also, we tried some different schedules: $t^\alpha\cdot(1-t)^\alpha$, where $\alpha=2,3,4$, while keep the magnitude as $\sigma^2(x_t)$. However, these different noise schedules yield almost the same performance, proving that FMA is not sensitive to the noise schedule. On the other hand, we found that FMA is very sensitive to the strength. We tried several strengths of the noise in the form of $k\cdot t \cdot (1-t)$, where $k$ is the corresponding noise magnitude. As we can see, if the noise magnitude is too large, the performance decreases dramatically.
>
> |Method | Aircraft | SAT | DTD | SUN | UCF
> |-|-|-|-|-|-|
> |CLIP|24.8|47.8|43.8|62.5|66.7|49.1|
> |k=$\sigma^2(x_t)$|**46.4**|**87.9**|**72.0**|73.3|**83.1**|
> |k=2$\sigma^2(x_t)$|45.8|87.2|71.1|**83.2**|71.2|
> |k=5$\sigma^2(x_t)$|43.6|83.8|69.1|79.5|68.8|71.7|
> |k=1|37.9|78.1|64.2|74.2|54.1|71.8|
>
>
> 3. Benefit from noise. Adding noise can enrich the training trajectories, thus can potentially boost the FMA's performance. This point is supported by the ablation study 6c. However, adding noise usually makes it more difficult to train the velocity.     For more details, check the paper [1]
>
> [1] Zhu, Kaizhen, et al. "Diffusion Bridge or Flow Matching?" ICLR(2025).

---

> ### Author Response · Authors · 2025-11-26
> **Response to Reviewer CrjH(2/3)**
>
> > Q3: Did you try higher-order or adaptive ODE solvers (Heun/RK) to reduce truncation error and improve margins at the same step budget?
>
> Yes, actually, we have tried several higher-order ODE solvers, including Heun/RK.  However, there seems to be no advantages over the Euler method.  To demonstrate this, we give the result on difficult datasets using Heun/RK in the inference stage as follows (the experiment setting is: CLIP + FMA, 16-shot, ViT-B/16)
>
> |Method | Aircraft | SAT | DTD | SUN | UCF | Avg|
> |-|-|-|-|-|-|-|
> |CLIP|24.8|47.8|43.8|62.5|66.7|49.1|
> |+FMA(Euler)|46.4|87.9|72.0|73.3|83.1|72.5
> |+FMA(Heun)|46.4|87.7|71.8|73.4|83.1|72.5
> |+FMA(RK)|46.4|87.7|71.9|73.3|83.1|72.5
>
> We can see that the performance of these three different ODE solvers yields almost the same results.
> Reason: Because of the limited data and the coupling enhancement strategy, all training trajectories can be viewed as non-crossing in a high-dimensional manifold. Therefore, it is very likely to learn a highly linearized transport map, where different ODE solvers will yield similar results.
>
>
> > Q4: Is fixed pairing too rigid for multi-modal classes? Would transporting toward multiple positive prototypes or a class subspace help?
>
>
> This is really an insightful suggestion.  First, for the fixed pairing,  we added a theoretical section (See Section 4 and Appendix A) to prove why this fixed pairing can work. On the other hand, while we agree that real-world image distributions are multi-modal, modeling a complex target (like a class subspace or multiple prototypes) is extremely challenging in the few-shot setting.  In this data-scarce regime, the flexibility of dynamic prototypes can also introduce instability and optimization difficulties.
>
> This point is also supported in the experimental results (Table 3). In Table 3, the CoCoOp baseline is exactly a strategy to model multiple positive prototypes. Specifically, given an image, the CoCoOp model will generate image-specific text embeddings. Thus, when training FMA based on CoCoOp, we are transporting images of the same class toward multiple positive prototypes. However, FMA's performance is not significantly better than other baselines.
>
>
>
>
>
>
> > Q5: Have you considered adding a discriminative loss on intermediate states (e.g., contrastive/margin) to align transport with classification throughout the trajectory?
>
> Yes, we have considered adding a contrastive loss during the training,  similar to the contrastive flow matching [1]. Specifically,  we first use the velocity $u^\theta _ t(x _ t)$ to obtain a predicted text feature: $\hat{x} _ 1 = x _ t + (1-t) \cdot u^\theta _ t(x _ t)$ . Then we use $\hat{x} _ 1$ to calculate a cross-entropy loss $\mathcal{L} _ {CE} = CE(\hat{x} _ 1, \mathrm{label})$,  just like what a typical classification model does.  Intuitively, $\mathcal{L} _ {CE}$ serves as a contrastive target, requiring the velocity to move far from text embeddings of different classes. Therefore, the final training objective is: $\mathcal{L} = \lambda\mathcal{L} _ {FM} + (1-\lambda)\mathcal{L} _{ CE}$:
>
> |Method | Aircraft | SAT | DTD | SUN | UCF
> |-|-|-|-|-|-|
> |CLIP|24.8|47.8|43.8|62.5|66.7|49.1|
> |$\lambda=1$|**46.4**|**87.9**|**72.0**|**73.3**|**83.1**|
> |$\lambda=0.5$|45.7|87.3|71.5|72.7|82.1|
> |$\lambda=0.3$|44.3|85.2|70.2|71.4|80.9|
> |$\lambda=0$|44.1|82.8|68.5|70.3|77.9|
>
>
> As observed, adding a discriminative objective does not improve performance. **Reason:** The discriminative objective **conflicts** with our coupling enforcement strategy. While the coupling strategy strictly enforces the transport direction along a straight line towards the correct class embedding, the discriminative objective forces the vector field to deviate from this straight path to satisfy local margin constraints, which harms the quality of the velocity field.
>
> [1]Stoica, George, et al. "Contrastive Flow Matching." CVPR(2025).

---

> ### Author Response · Authors · 2025-11-26
> **Response to Reviewer CrjH(3/3)**
>
> > Q6:  Can you report detailed overhead (velocity net size, train/infer time, average ESS steps) and scaling with the number of classes?
>
> We appreciate this practical question regarding efficiency. To address this, we conducted a comprehensive comparison of computational costs between FMA and five baseline methods.
>
> **Experimental Setup:** For a fair comparison, we fixed the training/test batch size at 32 and measured the average training time per iteration and peak memory usage on a single NVIDIA RTX 3090 GPU. The results are summarized below:
>
> |Method|Training Time |Training Memory|Inference time/(CLIP inference time)|
> |-|-|-|-|
> |CLIP|-|-|1 (10.2ms)|
> |CoOp|712ms|2770MB|1.2|
> |CoCoOp|1912ms|12318MB|3.2|
> |CLIP-Adapter|157ms|1376MB|1.0|
> |CLIP-LoRA|331ms|5406MB|1.3
> |FMA|198ms |2842MB|1.2 ($M$=1)
>
>
> **Analysis & Conclusions:**
>
> 1.  **High Training Efficiency:** As shown in the table, FMA is **significantly faster** than prompt-tuning methods (e.g., ~3.5x faster than CoOp) and comparable to the most efficient Adapter-based methods. This efficiency stems from our flow matching objective, which does not require backpropagation through the heavy text encoder during training.
>
> 2.  **Flexible Inference Trade-off:** We acknowledge that multi-step inference (M>1) increases latency. However, FMA offers a flexible **trade-off between speed and accuracy**.  For example, the performance of CLIP+FMA on EuroSAT, varying different $M$ is as follows.  **M=1** (which has a similar inference cost to CoOp) already achieves a massive improvement over Zero-Shot (47.8->73.6):
>
> |Steps|0(Zero-Shot)|1|2|3|4|5|
> |-|-|-|-|-|-|-|
> |Acc|47.8|73.6|83.1|85.8|87.0|87.5|
>
>
> 3.  **Contextualizing Efficiency:** In the few-shot learning community, complex models like CoCoOp (which is ~3x slower in inference) are widely accepted because they solve critical issues like generalization. Similarly, FMA tackles the difficult "entangled distribution" problem. Given its superior training speed and flexible inference steps, we believe FMA presents a highly competitive solution.
>
> **Average ESS Steps:**
> |Baseline| CLIP|CoOp|CoCoOp|CLIP-Adapter|CLIP-LoRA|
> |-|-|-|-|-|-|
> |Average $M$|3.7|1.7|4.2|3.1|2.2|
>
> For more details of $M$ on a specific dataset, see Appendix C.
>
> **Scaling with number of Classes**
> Similar to other PEFT methods, the computational complexity of FMA is $O(KN)$. Here $N$ is the number of classes.
>
>
> > W1: Lack of formal guarantees for coupling assumptions, reliance on validation to set inference steps, missing comparisons with higher-order ODE solvers, coarse difficulty metric, and limited analysis of compute trade-offs and failure modes.
>
> For a formal guarantee, we have added a section in the revised manuscript to provide a theoretical background for FMA. Generally, the theory can be summarized as follows:
> - The target of FMA is to find a marginal velocity $v_t(x_t)$, which can meet the following two requirements: 1) convert image features into text features; **2) make sure the transferred text features are the correct categories of the starting image features.** Because canonical flow matching cannot guarantee 2), we need to design a new framework, namely FMA.
> - We proved that the Coupling Enhancement is equivalent to defining the conditional probability path $p_t(x_t|z_c)$, when $t=0$.
> - Similar with canonical flow matching, we define $p_t(x_t|z_c)$ as linear interpolation when $t \neq 0$, then prove under a certain definition, $v_t(x_t)$can meet requirement 1.
> - Under Coupling Enhancement, we prove that $v_t(x_t)= v_t(x_t|z_c)$ in theory, where $z_c$ is the corresponding text embedding of $x_t$.
> - Using $v_t(x_t)= v_t(x_t|z_c)$, we prove $v_t(x_t)$ will meet requirement 2, which means it can guarantee that the transferred feature belongs to the same category as the initial image feature.
>
>
> Other weaknesses are answered in the corresponding questions.

---

### Official Review · Reviewer_BSTW · 2025-10-30

**Soundness:** 3
**Presentation:** 2
**Contribution:** 2
**Rating:** 4
**Confidence:** 4

**Summary:**

This paper addresses the challenge of achieving precise cross-modal alignment in vision-language models (VLMs) for few-shot learning. The authors argue that existing parameter-efficient fine-tuning (PEFT) methods—such as prompt tuning, adapter-based, and LoRA-based approaches—perform only a "one-step" adjustment of features, which is insufficient for complex datasets where modalities are highly entangled. To overcome this limitation, the authors propose Flow Matching Alignment (FMA), a model-agnostic framework that leverages flow matching theory to enable multi-step feature transformation. FMA incorporates three key designs: coupling enforcement to preserve class correspondence, noise augmentation to mitigate data scarcity, and an early-stopping solver for efficient and accurate inference. Extensive experiments on 11 benchmarks show that FMA consistently improves performance, especially on challenging datasets, and integrates seamlessly with various backbones and PEFT methods.

**Strengths:**

1. Novel application of flow matching to cross-modal alignment in few-shot learning, moving beyond generative tasks.
2. Effective design choices (e.g., early-stopping solver, noise augmentation) that address practical challenges in training and inference.

**Weaknesses:**

1. No analysis of computational overhead or inference latency introduced by multi-step transformation.
2. Ablation studies do not explore the sensitivity of performance to hyperparameters like inference steps.
3. The early-stopping strategy uses a fixed step count rather than a sample-adaptive criterion, which may limit optimality.

**Questions:**

1. Could the author provide more detailed information across different datasets in section 4.2 GENERALIZATION ABILITY, where only average performance was given?
2. Was any exploration done into adaptive early-stopping criteria (e.g., based on feature discriminability) rather than a fixed stepsize?
3. How does FMA perform in cross-modal retrieval or other downstream tasks beyond classification, given its alignment-focused design?
4. Could the authors provide more intuition or theoretical insight into why coupling enforcement preserves class-level correspondence in high-dimensional feature spaces?

---

> ### Author Response · Authors · 2025-11-26
> **Response to Reviewer BSTW (1/2)**
>
> We thank the reviewer for their constructive assessment and the opportunity to clarify the details of our approach. Our responses to the specific questions are as follows.
>
> > Q1:  Could the author provide more detailed information across different datasets in section 4.2 GENERALIZATION ABILITY, where only average performance was given?
>
> We have added related details in Appendix E. Besides, we added four other baselines to demonstrate the generalization ability of FMA.
>
>
> > Q2 && W3:  Was any exploration done into adaptive early-stopping criteria (e.g., based on feature discriminability) rather than a fixed stepsize?
>
> In this paper, we rely on a validation set to determine the inference steps M strictly to **follow the standard evaluation protocol in Few-Shot Learning** (similar to how Tip-Adapter/CLIP-Adapter tunes αα on validation sets). Besides, we agree that adaptive stopping has the potential to obtain higher. However, most adaptive strategies still rely on some hyperparameters.  For example, we can feature discriminability as a criterion.
>
> Given an image feature, first calculate the logits $L_t$ at timestep $t$. If $\mathrm{Top1}(L_t) - \mathrm{Top2}(L_t) > \epsilon$, then we can conclude that the current feature is discriminative enough to give a decision, thus stop the inference at the current timestep.
>  However, it is difficult to determine the value of $\epsilon$, and for different values of $\epsilon$, related results are as follows (16-shot, ViT-B/16, CLIP+FMA). For different datasets, we need different $\epsilon$ to obtain better results. **In this situation, $\epsilon$  is another hyper-parameter, just like $M$, and we still need validation tuning for this strategy.**
>
> |Method | Aircraft | SAT | DTD | SUN | UCF | Avg|
> |-|-|-|-|-|-|-|
> |CLIP|24.8|47.8|43.8|62.5|66.7|49.1|
> |ESS|**46.4**|**87.9**|**72.0**|**73.3**|**83.1**|**72.5**
> |$\epsilon$=0.001|42.1|86.9|**72.0**|71.9|**83.1**|71.2|
> |$\epsilon$=0.0007|46.0|86.1|71.3|72.4|82.7|71.7|
> |$\epsilon$=0.0005|**46.4**|**87.9**|70.2|**73.3**|81.5|71.8|
>
> Theoretically,  if we can train a perfect velocity, then the flow toward $t=1$ will always work.  However, due to the limited data and highly entangled feature space, the trained velocity is usually imperfect. Meanwhile,  there is still no tool to analyze how imperfect a velocity is in the flow matching theory. Therefore, tuning on a validation set is the most practical method.
>
>
> > Q3:  How does FMA perform in cross-modal retrieval or other downstream tasks beyond classification, given its alignment-focused design?
>
> We clarify that while FMA is designed for alignment, it is currently tailored specifically for **classification tasks** (mapping Image → Text) and cannot be directly applied to cross-modal retrieval in its current form.
>
> **Reason:** Cross-modal retrieval typically requires a shared metric space for bidirectional queries (Image → Text and Text → Image). However, our FMA is designed as a unidirectional transformation process. We train the velocity field explicitly to transport image features towards text features to improve classification accuracy. While ODEs are theoretically invertible, our training objective (mapping complex image distributions to discrete text prototypes) makes the reverse generation (Text → Image) ill-posed without additional generative constraints.
>
> Therefore, applying FMA to retrieval or generation would require modifying the training objective (e.g., training a bidirectional flow), which we consider a promising direction for future work.
>
>
>
> >Q4: Could the authors provide more intuition or theoretical insight into why coupling enforcement preserves class-level correspondence?
>
>
> For a formal guarantee, we have added a section in the revised manuscript to provide a theoretical background for FMA. Generally, the theory can be summarized as follows: (For more details, see Section 4 and Appendix A.)
> - The target of FMA is to find a marginal velocity $v_t(x_t)$, which can meet the following two requirements: 1) convert image features into text features; **2) make sure the transferred text features are the correct categories of the starting image features.** Because canonical flow matching cannot guarantee 2), we need to design a new framework, namely FMA.
> - We proved that the Coupling Enhancement is equivalent to defining the conditional probability path $p_t(x_t|z_c)$, when $t=0$.
> - Similar with canonical flow matching, we define $p_t(x_t|z_c)$ as linear interpolation when $t \neq 0$, then prove under a certain definition, $v_t(x_t)$can meet requirement 1.
> - Under Coupling Enhancement, we prove that $v_t(x_t)= v_t(x_t|z_c)$ in theory, where $z_c$ is the corresponding text embedding of $x_t$.
> - Using $v_t(x_t)= v_t(x_t|z_c)$, we prove $v_t(x_t)$ will meet requirement 2, which means it can guarantee that the transferred feature belongs to the same category as the initial image feature.
>
> For more details, see Section 4 and Appendix A.

---

> ### Author Response · Authors · 2025-11-26
> **Response to Reviewer BSTW (2/2)**
>
> > W1: No analysis of computational overhead or inference latency introduced by multi-step transformation.
>
> To address this, we conducted a comprehensive comparison of computational costs between FMA and five baseline methods.
>
> **Experimental Setup:** For a fair comparison, we fixed the training/test batch size at 32 and measured the average training time per iteration and peak memory usage on a single NVIDIA RTX 3090 GPU. The results are summarized below:
>
> |Method|Training Time |Training Memory|Inference time/(CLIP inference time)|
> |-|-|-|-|
> |CLIP|-|-|1 (10.2ms)|
> |CoOp|712ms|2770MB|1.2|
> |CoCoOp|1912ms|12318MB|3.2|
> |CLIP-Adapter|157ms|1376MB|1.0|
> |CLIP-LoRA|331ms|5406MB|1.3
> |FMA|198ms |2842MB|1.2 ($M$=1)
>
>
> **Analysis & Conclusions:**
>
> 1.  **High Training Efficiency:** As shown in the table, FMA is **significantly faster** than prompt-tuning methods (e.g., ~3.5x faster than CoOp) and comparable to the most efficient Adapter-based methods. This efficiency stems from our flow matching objective, which does not require backpropagation through the heavy text encoder during training.
>
> 2.  **Flexible Inference Trade-off:** We acknowledge that multi-step inference (M>1) increases latency. However, FMA offers a flexible **trade-off between speed and accuracy**.  For example, the performance of CLIP+FMA on EuroSAT varying different $M$ is as follow.  **M=1** (which has a similar inference cost to CoOp) already achieves a massive improvement over Zero-Shot (47.8->73.6):
>
> |Steps|0(Zero-Shot)|1|2|3|4|5|
> |-|-|-|-|-|-|-|
> |Acc|47.8|73.6|83.1|85.8|87.0|87.5|
>
>
> 3.  **Contextualizing Efficiency:** In the few-shot learning community, complex models like CoCoOp (which is ~3x slower in inference) are widely accepted because they solve critical issues like generalization. Similarly, FMA tackles the difficult "entangled distribution" problem. Given its superior training speed and flexible inference steps, we believe FMA presents a highly competitive solution.
>
> >  W2: Ablation studies do not explore the sensitivity of performance to hyperparameters like inference steps.
>
>
>
> **Regarding Inference Steps:** We respectfully clarify that the sensitivity to inference steps (M) was analyzed in the original manuscript. As shown in **Figure 6(b)**, we evaluated the performance with M ranging from 1 to 10, demonstrating that our Early Stopping Solver (ESS) outperforms the vanilla solver across different step budgets.
>
> **Additional Ablation Studies:** However, we agree that a broader exploration of other hyperparameters strengthens the paper. Therefore, we have added **Appendix D** in the revised manuscript to include comprehensive ablations on: Noise Strength,  ODE Solvers and  Loss Objectives.

---

### Official Review · Reviewer_2Lin · 2025-10-31

**Soundness:** 2
**Presentation:** 3
**Contribution:** 2
**Rating:** 2
**Confidence:** 3

**Summary:**

This paper proposes a method to improve alignment performance between modalities in cross-modal models.
It claims that existing methods fail to align well on challenging datasets because they attempt one-step alignment,
and proposes a multi-step approach to align the embedding vectors of the two modalities.
Specifically, it performs flow matching to transform from images to the distribution of text embedding vectors.

**Strengths:**

- This paper is well-written and well-structured.
- Alignment of embedding vectors across multi modalities is an important research topic.
- Flow-matching alignment seems novel. However, its necessity is questionable, and it might be just a combination of new techniques.
- In experiments, the proposed method outperforms baselines on class classification tasks. However, as written in Weaknesses, it is unclear whether the evaluation is well-designed to confirm the claims.

**Weaknesses:**

- The motivation for multi-step adjustment is unclear.
First, the definition of one-step adjustmentfor poor performance in existing methods is ambiguous.
For example, is the claim that PEFT's optimization objective function is inappropriate, or that optimization is insufficient due to difficult learning?
Figure 2 discusses PEFT's characteristics compared to LP, but the validity of using LP as a baseline for this discussion is unclear.
It is unclear how this connects to the statement: “these methods try to adjust their general aligned multi-modal distribution towards the golden distribution by one rectification step.”

- The experimental setups are insufficiently described, resulting in a lack of reproducibility.
For example, it states that velocity networks are learned, but I could not find a description of the specific structure of the velocity networks.
There is no definition of $\sigma^2(\cdot)$. There seems also no report of the number of steps M for the proposed method across each dataset.
In addition, there is no evaluation of statistical significance.

- The baseline varies depending on the evaluation.
While Table 1 compares against 8 baselines, Table 2 has one baseline and Table 3 has five.
Specifically, the baseline compared in Table 2 is one of the weaker baselines among those appearing in Table 1.
Although there are practical limitations on the number of experiments, comparing against the strongest baseline yields more convincing results.

- The proposed method seems computationally expensive.
It requires preparing velocity networks and performing multiple updates during inference (Algorithm 2).
How does the computational cost compare to the CLIP-Adapter with two linear layers? How does it compare to PEFT?
Since the performance improvement over CLIP-LoRA is only 0–2%, the heavy inference cost makes the proposed method less useful.

- Minor issues
- The space is filled, making it difficult to read.
 The absence of a single line of space before and after figures and tables, such as the caption for Figure 4, violates the template.

**Questions:**

- What is the definition of one-step adjustment? Does it mean that the objective function is set once or that the optimization is only one step? Are Fig. 1(b)-(d) optimal embeddings in some sense?

- In Fig. 2, isn't it a bit simplistic to conclude that PEFT is weak on challenging datasets based on LP?
Couldn't one also conclude that LP is strong on more challenging datasets?

- What happens if stronger methods are used as baselines in Table 2? Also, did you check the standard deviation of the results and their statistical significance?

- How about the comparison of computational cost?

---

> ### Author Response · Authors · 2025-11-26
> **Response to Reviewer 2Lin (1/3)**
>
> We thank the reviewer for the constructive feedback and the time spent reviewing our work. We have addressed the concerns point-by-point below.
>
> > Q1: What is the definition of a one-step adjustment? Does it mean that the objective function is set once or that the optimization is only one step?
>
>  The term "one-step adjustment" is: During the **inference** stage, the input initial image features are transferred by a single forward pass of the trained model.  This definition is mentioned in Lines 108-109.
>
> By definition, we can conclude that while discussing "one-step" or "multi-step" methods, we refer specifically to the **inference procedure**, instead of the optimization procedure.  Therefore, the "one-step adjustment" doesn't mean the objective function is set once, or the optimization is only one step.  To elaborate, we can compare FMA with existing PEFT methods as follows:
>
> -   Existing PEFT (One-Step):  Methods like Adapters, Prompt-Tuning, and LoRA produce the adapted feature via a single forward pass.  Geometrically, this is equivalent to applying a  displacement vectore $\Delta(x_{ori})$ to the oiginal feature$x_{ori}$ in a single step, i.e., $x_{new} = x_{ori} + \Delta(x_{ori})$ . Also, $\Delta(x_{ori})$ is noted as the black arrow in Figure 1(bcd).  As we can see,  the model must predict the final target feature $x_{new}$ **directly** from $x_{ori}$.  which is challenging when the underlying cross-modal distribution is complicated or entangled.
>
> -   FMA (Multi-Step):  In contrast, FMA models the adaptation as an ODE: $dx_t/dt=v_t​(x_t)$. During inference, we integrate this velocity field over time. This allows the model to break down the whole complex transformation into a multi-step adaptation. At each step, FMA only needs to perform a local update based on the current state, which makes it easier to model a complex transformation.
>
> This motivation is similar to image generation using Flow Matching. As you know, previous image generation methods usually try to model the complex transformation (noise to image) directly in one step, thus struggling with the generation quality. To solve this,  flow matching instead applies a multi-step strategy. At each step, the velocity will move the noise towards the direction of the mages. In this paper, we found that PEFT is difficult to learn a complicated transformation; therefore, we propose FMA to learn this transformation in a multi-step manner.
>
>
>
> >Q1: Are Fig. 1(b)-(d) optimal embeddings in some sense?
>
> These embeddings visualize the feature distributions at the **converged state** of the respective models. Specifically, we train each PEFT method until the loss function stabilizes, and then transform the initial features to obtain the final representations shown in Fig. 1(b)-(d). Therefore, they represent the **optimized embeddings** with respect to their training objectives, reflecting the best alignment capability achievable by these methods under the standard training protocol.
>
>
>
>
> > Q2:   In Fig. 2, isn't it a bit simplistic to conclude that PEFT is weak on challenging datasets based on LP? Couldn't one also conclude that LP is strong on more challenging datasets?
>
> We would like to clarify that our claim "PEFT is weak on challenging datasets" specifically refers to its **cross-modal alignment capability**, rather than its absolute accuracy.
>
>
> Our logic is based on the widely accepted understanding of Linear Probing (LP):
>
> 1.  **Source of Gains:** The improvement over Zero-Shot comes from two factors: (a) the extra information from new training examples, and (b) the alignment capability of the algorithm itself.
>
> 2.  **LP's Limitation:**  We admit that LP may be strong on challenging datasets, which means LP may show **strong absolute performance**. However,  this gain predominantly stems from data access rather than alignment. **It is widely acknowledged that LP has weak alignment capability** because it is restricted to learning a simple linear transformation.
>
> 3.  **Conclusion:** Therefore, we use LP as a baseline to factor out the data's contribution. If PEFT methods yield only marginal improvements over LP (as shown in Fig. 2), it implies that their cross-modal alignment capability is insufficient.
>
> We have revised the manuscript for better illustration(Lines 90-96).
>
> > Q3: stronger baselines in Table 2?  Standard deviation and its statistical significance.
>
>
> To address your concern, we have expanded Table 2 by incorporating **four additional stronger baselines**. We observe that even against these stronger methods, FMA consistently yields significant performance improvements.
>
> Regarding statistical reliability, all results reported in the main paper are averaged over **three independent experiments** with random seeds. We have explicitly analyzed the standard deviations and calculated statistical significance (p-values) to ensure the validity of our conclusions. These detailed statistics are provided in **Appendix C**.

---

> ### Author Response · Authors · 2025-11-26
> **Response to Reviewer 2Lin (2/3)**
>
> > Q4:  How about the comparison of computational cost?
>
>
> We appreciate this practical question regarding efficiency. To address this, we conducted a comprehensive comparison of computational costs between FMA and five baseline methods.
>
> **Experimental Setup:** For a fair comparison, we fixed the training/test batch size at 32 and measured the average training time per iteration and peak memory usage on a single NVIDIA RTX 3090 GPU. The results are summarized below:
>
> |Method|Training Time |Training Memory|Inference time/(CLIP inference time)|
> |-|-|-|-|
> |CLIP|-|-|1 (10.2ms)|
> |CoOp|712ms|2770MB|1.2|
> |CoCoOp|1912ms|12318MB|3.2|
> |CLIP-Adapter|157ms|1376MB|1.0|
> |CLIP-LoRA|331ms|5406MB|1.3
> |FMA|198ms |2842MB|1.2 ($M$=1)
>
>
> **Analysis & Conclusions:**
>
> 1.  **High Training Efficiency:** As shown in the table, FMA is **significantly faster** than prompt-tuning methods (e.g., ~3.5x faster than CoOp) and comparable to the most efficient Adapter-based methods. This efficiency stems from our flow matching objective, which does not require backpropagation through the heavy text encoder during training.
>
> 2.  **Flexible Inference Trade-off:** We acknowledge that multi-step inference (M>1) increases latency. However, FMA offers a flexible **trade-off between speed and accuracy**.  For example, the performance of CLIP+FMA on EuroSAT varying different $M$ is as follows.  **M=1** (which has a similar inference cost to CoOp) already achieves a massive improvement over Zero-Shot (47.8->73.6):
>
> |Steps|0(Zero-Shot)|1|2|3|4|5|
> |-|-|-|-|-|-|-|
> |Acc|47.8|73.6|83.1|85.8|87.0|87.5|
>
>
> 3.  **Contextualizing Efficiency:** In the few-shot learning community, complex models like CoCoOp (which is ~3x slower in inference) are widely accepted because they solve critical issues like generalization. Similarly, FMA tackles the difficult "entangled distribution" problem. Given its superior training speed and flexible inference steps, we believe FMA presents a highly competitive solution.
>
> >W1: For poor performance of PEFT, is the claim that PEFT's optimization objective function is inappropriate, or that optimization is insufficient due to difficult learning?
>
> It is neither.  Our conclusion is:  The limitation of PEFT lies in its "one-step" inference mechanism.
>
> We contend that the intrinsic nature of existing PEFT methods is a single-step transformation (i.e., $x_{new}​=x_{ori}​+\Delta(x_{ori})$).  This "one-step" property forces the model to learn the transformation only based on the initial input.  However, when the feature space is highly entangled (typical of "Difficult" datasets),  learning such a complex transformation directly is inherently challenging.
>
> Regarding your hypotheses (inappropriate objective or difficult learning), we validated our setup by strictly following the **standard protocols** of the official implementations. Our reproduced results align closely with the reported figures in the original papers. This consistency suggests that the suboptimal performance stems from the inherent limitations of the method itself, rather than implementation or optimization issues.
>
>
>
>
>
>
> > W1: The validity of using LP as a baseline for this discussion is unclear. It is unclear how this connects to the following statement.”
>
> Thanks for the comment. As we responded in Q2, the rationale for comparing PEFT with LP is to factor out the data contribution and measure the algorithm's cross-modal alignment capability.  The following statement is our explanation about why PEFT's alignment ability is weak on challenging datasets: a one-step transformation is difficult when the cross-modal distribution is highly entangled. This may be a little misleading, and we have revised the manuscript for better explanation (Lines 106-113).

---

> ### Author Response · Authors · 2025-11-26
> **Response to Reviewer 2Lin (3/3)**
>
> > W2:  Reproducibility, Experiment setup, velocity network,  definition of $\sigma^2(x_t)$, $M$ for different datasets, statistical significance.
>
> Thanks for the advice. We have added the training details in the revised paper (Lines 392-395).  The training epoch or batch size of FMA is the same as its baseline. The architecture of the velocity network is the same as MAR[1]. By default, there are 12 ResNet blocks, and all hidden dimensions equal CLIP’s feature dimension.
>
> $\sigma(x^2)$ in $\hat{x_t} \sim\mathcal{N}(\hat{x_t}|x_t,t\cdot(1-t)\cdot\sigma(x_t^2))$. Here $\sigma(x_t^2)\in\mathbb{R}$  is the standard deviation of $x_t\in \mathbb{R}^d$: $\sigma(x^2_t)=\sqrt{\sum_{i=1}^d(x_t^i-\mu(x_t))^2} /d$, where $\mu(x_t) =\sum_{i=1}^dx^i_t/d$ is the mean of $x_t$. $x^i_t\in \mathbb{R}$ is the i-th dimension of $x_t$.
>
> For $M$, we report the number of steps in Appendix C.
>
> **To reproduce our results, we have released the code of our project:  https://anonymous.4open.science/r/FMA-1F78**
>
>
> [1]Li, Tianhong, et al. "Autoregressive image generation without vector quantization." Advances in Neural Information Processing Systems 37 (2024)
>
>
> > W3 && W4: The baseline evaluation is non-convincing and seems computationally expensive.
>
> Thanks for the concern. These two weaknesses are answered in  Q3 && Q4.
>
>
> > W5: The space is filled, making it difficult to read.
>
> We thank the reviewer for pointing out this oversight. We have revised the manuscript, strictly following the requirements of the template.

---

> > ### Comment · Reviewer_2Lin · 2025-11-28
> > **Thank you for your feedback!**
> >
> > I am grateful for the new results you have provided in response to my comments. While your answers have clarified the paper, they have also raised new concerns.
> >
> > - I have understood that one step adjustment means one-step inference.
> > However, if we assume the model possesses universal approximation capability, could it not learn one-step alignment under an appropriate objective function?
> > I understand that Fig. 1(b)-(d) shows inference results after learning.
> > Does this imply that while an ideal function minimising the objective function can achieve alignment, an empirically obtained model cannot?
> > Are there any clear causes, e.g., one-step adjustment is difficult to design an objective function, difficult to optimise, requires a large sample size, or is difficult to generalise?
> >
> > - Seeing the newly reported table about the setting of M, M=1 often becomes the optimal. Is the claim that multi-step adjustment is crucial correct?
> >
> > - Comparing the inference time of M=1 with that of the baselines is unfair because multi-step adjustment is the core of the proposed method. The inference time at the optimal M should be reported.
> > If statistical significance is not consistently demonstrated when compared to CLIP-LoRA, it remains unclear whether incurring M times the inference cost is beneficial.

---

> > > ### Author Response · Authors · 2025-11-28
> > > **Follow up (1/2)**
> > >
> > > Thanks for the response! We address your new concerns point-by-point below.
> > > > However, if we assume the model possesses universal approximation capability, could it not learn one-step alignment under an appropriate objective function?  I understand ....
> > >
> > >
> > >
> > > We agree that theoretically, given the universal approximation capability, a one-step model can achieve this alignment.  But as you mentioned, empirically obtained one-step models cannot.  There are two reasons: theory and practice.
> > >
> > >
> > >  There are several potential reasons for this, such as suboptimal training objectives, limited network capacity, or insufficient data. While the first two are hard to evaluate, **data scarcity** is clearly a bottleneck in few-shot learning, where usually fewer than 16 examples are available per class.
> > >
> > > Another reason is the complicated cross-modal distribution (for difficult datasets). This means the optimal alignment function is very complex to learn. In this situation, it is even more difficult for one-step approaches to learn, which directly predict the output based on the input. This challenge is similar to that in image generation: (1)
> > > Theoretically, one-step methods like VAEs and GANs, which directly map noise into images, can learn a good generative model.  In practice, however, they often suffer from training instability (e.g., mode collapse) or suboptimal generation quality. In contrast, Flow Matching models achieve superior results by **decomposing the complex transport trajectory into multiple steps**. At each step, the model only needs to learn a **local update** based on the current state, which is significantly easier to optimize and generalize. FMA brings this same advantage to few-shot alignment.
> > >
> > >
> > > >  Seeing the newly reported table about the setting of M, M=1 often becomes the optimal. Is the claim that multi-step adjustment is crucial correct?
> > >
> > >
> > > We appreciate this opportunity to clarify the scope of our contribution. We respectfully point out that our claim regarding the necessity of multi-step adjustment is specifically targeted at **"Difficult" datasets**, where the cross-modal distribution is highly entangled, and the ideal transformation is complex (as stated in Lines 110-112).
> > >
> > > 1.  **On Easy Datasets:** We acknowledge that for datasets with simpler distributions, one-step methods are often sufficient. Therefore, you can see that the optimal M frequently converges to 1.
> > >
> > > 2.  **On Difficult Datasets:** However, for challenging tasks, multi-step adjustment is crucial. As shown in the table below, the optimal inference steps (M) are consistently greater than 1 (ranging from 2 to 9) on all difficult datasets. This empirical evidence strongly validates that a single rectification step is insufficient for disentangling complex cross-modal manifolds.
> > >
> > > *Note*: We also want to clarify that dataset difficulty is defined by **Zero-Shot CLIP performance**.  FMA performs better on difficult tasks, as concluded by the experiments, and these conclusions exactly match our motivation:
> > > Lower Zero-Shot accuracy → Complicated cross-modal distribution → Ideal transformation is difficult to learn. Therefore, PEFT's one-step improvement over LP is marginal, and we need FMA's   **multi-step capability** to learn this difficult transformation.
> > > |Baseline| FGVC |EuroSAT| DTD |SUN |UCF|
> > > |-|-|-|-|-|-|
> > > |CLIP |7| 6| 5| 3| 4|
> > > |CoOp |2 |2 |2 |3 |4|
> > > |CoCoOp| 7| 2| 6| 6| 9|
> > > |CLIP-Adapter |3 |7 |5 |3 |2|
> > > |CLIP-LoRA |3 |5 |2 |3 |2|

---

> ### Author Response · Authors · 2025-11-28
> **Follow up (2/2)**
>
> >Comparing the inference time of M=1 with that of the baselines is unfair because multi-step adjustment is the core of the proposed method. The inference time at the optimal M should be reported.
>
>
> We initially reported M=1 because the optimal steps vary across datasets.  And we agree that comparing the inference time at the optimal M is more appropriate. Since FMA targets challenging tasks, we calculated the average optimal step **on the difficult datasets**, which yields **M=4.2**. The updated table is below.
>
>
> |Method|Training Time |Training Memory|Inference time/(CLIP inference time)|
> |-|-|-|-|
> |CLIP|-|-|1 (10.2ms)|
> |CoOp|712ms|2770MB|1.2|
> |CoCoOp|1912ms|12318MB|3.2|
> |CLIP-Adapter|157ms|1376MB|1.0|
> |CLIP-LoRA|331ms|5406MB|1.3
> |FMA|198ms |2842MB|1.8 ($M$=4)
>
> **Analysis:**
>
> First, even with M=4, FMA (1.8x) remains significantly faster than CoCoOp(3.2x), a key baseline designed for few-shot learning.
>
> Additionally, while FMA is slightly slower than CoOp, CLIP-LoRA, and  CLIP-Adapter, this absolute latency increase (<10ms per batch on 3090) is negligible in most real-world scenarios. Crucially, when considered alongside FMA's other advantages—such as faster training, lower memory usage, and significantly higher performance **on difficult datasets**—we believe FMA offers a highly competitive solution.

---

### Author Response · Authors · 2025-11-26
**General Response to All Reviewers.**

We appreciate all suggestions and comments and carefully revise our paper accordingly. Our major revisions include the following four aspects:

1.  In the Introduction, we provided a clearer explanation of our motivation(Lines 90-100, 105-127).

2.  We added a section to give a comprehensive theoretical illustration of FMA(Lines 343-381).

3.  In the Experiments:

    -   We added more details of our methods to reproduce the results(Lines 394-394). Besides, we release our code: https://anonymous.4open.science/r/FMA-1F78
    -   We compared four more methods in the generalization experiment (Table 1).
    -    We added more ablation studies in the Appendix.

4.  In the Appendix:

    -   Appendix A shows related proof of the theoretical section.
    -   Appendix B shows the training and inference algorithms.
    -   Appendix C shows related statistical significance, standard deviation, and the value of inference step $M$  of  Architecture Agnostic Experiment (Table 3).
    -   Appendix D shows more ablation studies of FMA, like the strength of noise, adaptive solver, and so on.
    - Appendix E shows the details of the Generalization Experiment (Table 1).
    -   Appendix F shows the analysis of the computational cost of FMA.


Please note that we colorized (blue) the revisions in the new version of the paper.

---

### Meta-Review · Area_Chair_PqjD · 2025-12-26

**Summary:**

This paper presents flow matching alignment (FMA), a novel cross-modal alignment technique based on flow matching. Unlike previous work that aligns embeddings of two different modalities in a single step, FMA aligns them through multiple steps motivated by flow matching in generative models. The reviewers appreciated the new application of flow matching and identifying limitations of existing approaches recognized,  and recognized the versatility and efficacy of FMA, comprehensive experiments, and clarity of the paper. However, they raised multiple crucial concerns with computational complexity (All), lack of theoretical groundings (BSTW, CrjH, girA), missing analysis on the sensitivity to hyperparameters (BSTW, CrjH, girA), limited flexibility due to the fixed step count (BSTW, CrjH), comparisons with weak baselines in some tables (2Lin), and reproducibility issues due to the lack of details (2Lin). The authors addressed the major concerns by, in particular, extensive analysis on computational complexity, additional comparisons with latest work, an alternative technique to determine the step size dynamically, theoretical foundation of the proposed method, and more details of implementation. The AC considers most of the critical concerns have been successfully assuaged and thus recommends acceptance of the paper. The authors are strongly encouraged to add all the new results and discussions in the camera-ready version of the paper, and to elaborate more on why and how FMA's latency increase was marginal in practice.

**Reviewer Concerns:**

[Not fully addressed]
- Computaional complexity (All)
	- *The proposed method, FMA, is an iterative method and thus inevitably imposes additional complexity compared to existing single-step PEFT methods. the reviewers unanimously pointed out this potential issue on complexity.*
	- *The newly added experimental results demonstrate that the computation and memory imposed additionally by the proposed method are moderate--smaller than most of previous work. even in the multi-step adjustment setting, it still remained efficient than CoCoOp and the additional latency to the other previous work was negligible in practice.*
	- *The AC thus considers this concern has been addressed to some extent. However, the authors should more clearly illustrate the reason for the marginal latency increase even with multiple steps for alignment, as the rebuttal simply enumerate wall-clock times without sufficiently detailed analysis.*
- Limited flexibility due to the fixed step count (BSTW, CrjH)
	- *This has not been perfectly resolved, but seems to the AC not very significant regarding that hyperparameter setting using validation set is common in few-shot learning. Also, the authors further explore an alternative technique to determine the step size dynamically in the rebuttal.*

[Well assuaged]
- Reproducibility issues due to the lack of details (2Lin)
	- *This concern can be easily addressed by a single round of revision; the revised manuscript clearly illustrates such details. Moreover, an anonymized codebase has been provided during the rebuttal period. Hence, the AC considers that this issue has been well resolved.*
- Comparisons with weak baselines in some tables (2Lin)
	- *The authors provide additional results with more baselines in Table 2 and 3 of the revision.*
- Missing analysis on the sensitivity to hyperparameters (BSTW, CrjH, girA)
	- *This concern has been well addressed by additional experiments in the rebuttal.*
- Lack of theoretical groundings (BSTW, CrjH, girA)
	- *The revision includes theoretical background of the proposed method, which looks sufficiently solid.*

**Reviewer Scores:**

The paper initially received mixed scores: one reject (2, 2Lin), one borderline reject (4, BSTW), and two borderline accept (6, CrjH & girA). The AC expects that, if the discussion phase has continued, the negative reviewers (2Lin, BSTW) may increase their scores up to borderline accept since the rebuttal well assuaged most of their concerns. The positive reviewers (CrjH, girA) may also increase their scores or at least keep their original ratings for the same reason.

---

### Decision · Program_Chairs · 2026-01-26

Accept (Poster)